# Inhibition of AXL receptor tyrosine kinase enhances brown adipose tissue functionality in mice

Vissarion Efthymiou [1,2,11], Lianggong Ding[1,11], Miroslav Balaz[1,3], Wenfei Sun [1,4,5], Lucia Balazova[1,3], Leon G. Straub[1,6], Hua Dong [1,7], Eric Simon[8], Adhideb Ghosh[1], Aliki Perdikari[1], Svenja Keller[1,9], Umesh Ghoshdastider[1], Carla Horvath[1], Caroline Moser[1], Bradford Hamilton[10], Heike Neubauer[10] & Christian Wolfrum [1] ✉

The current obesity epidemic and high prevalence of metabolic diseases necessitate efficacious and safe treatments. Brown adipose tissue in this context is a promising target with the potential to increase energy expenditure, however no pharmacological treatments activating brown adipose tissue are currently available. Here, we identify AXL receptor tyrosine kinase as a regulator of adipose function. Pharmacological and genetic inhibition of AXL enhance thermogenic capacity of brown and white adipocytes, in vitro and in vivo. Mechanistically, these effects are mediated through inhibition of PI3K/AKT/PDE signaling pathway, resulting in induction of nuclear FOXO1 localization and increased intracellular cAMP levels via PDE3/4 inhibition and subsequent stimulation of the PKA-ATF2 pathway. In line with this, both constitutive *Axl* deletion as well as inducible adipocyte-specific *Axl* deletion protect animals from diet-induced obesity concomitant with increases in energy expenditure. Based on these data, we propose AXL receptor as a target for the treatment of obesity.

Obesity is considered a worldwide epidemic and has been associated with a variety of metabolic and other complications, such as type II diabetes, cardiovascular disease and several types of cancer, to name but a few. At present, the most efficient approach for a long-term and sustainable weight loss are several types of bariatric surgery, furthermore novel therapies are developed aiming at reducing food intake through glucagon-like peptide 1 (GLP1) and/or glucose-dependent insulinotropic polypeptide (GIP) receptor signaling[1–3]. However, there are currently no FDA-approved weight-loss treatments that target energy expense.

Brown adipose tissue (BAT) is a major thermogenic organ in mammals and rodents, due to its capacity to uncouple the production

[1]ETH Zürich – Swiss Federal Institute of Technology, Department of Health Sciences and Technology, Laboratory of Translational Nutrition Biology, Institute of Food, Nutrition and Health, Schwerzenbach, Switzerland. [2]Joslin Diabetes Center, Section of Integrative Physiology and Metabolism, Research Division, Harvard Medical School, Boston, MA, USA. [3]Laboratory of Cellular and Molecular Metabolism, Biomedical Research Center, Slovak Academy of Sciences, Bratislava, Slovakia. [4]Department of Bioengineering, Stanford University, Stanford, CA, USA. [5]Department of Molecular and Cellular Physiology, Stanford University School of Medicine, Stanford, CA, USA. [6]Institute of Child Nutrition, Max Rubner-Institut, Federal Research Institute of Nutrition and Food, Karlsruhe, Germany. [7]Institute for Stem Cell Biology and Regenerative Medicine, Stanford University School of Medicine, Stanford, CA, USA. [8]Department of Global Computational Biology and Digital Sciences, Boehringer Ingelheim Pharma GmbH & Co. KG, Biberach an der Riss, Germany. [9]Mechanisms of Inherited Kidney Diseases Group, Institute of Physiology, University of Zurich, 8057 Zurich, Switzerland. [10]Department of CardioMetabolic Diseases Research, Boehringer Ingelheim Pharma GmbH & Co. KG, Biberach an der Riss, Germany. [11]These authors contributed equally: Vissarion Efthymiou, Lianggong Ding. ✉e-mail: christian-wolfrum@ethz.ch

of ATP from oxidative phosphorylation, thus dissipating chemical energy in the form of heat[4,5]. Brown adipocytes do not exist only in the classic BAT depots (e.g. interscapular BAT or iBAT), but they can also be found within particular white adipose tissue (WAT) depots. These cells - functionally and morphologically similar to brown adipocytes – have been described as "beige" or "brite", whereas the phenomenon of the appearance of such cells within a white adipose depot is referred to as "browning" or "beiging" of WAT[6,7]. After the relatively recent discovery that BAT can be present in adult humans[8–12] - localized in the supraclavicular, paravertebral, axillary and deep neck regions - as well as the observation that "browning" of WAT can also occur in humans[13], the brown adipose organ started being considered a clinically relevant promising target to increase energy expenditure (EE).

Physiologically, β-adrenergic stimulation, typically induced by cold exposure or other stimuli that increase catecholamine levels (e.g., burning[13]) activates BAT[14]. Specifically, the second messenger cyclic AMP (cAMP) is responsible for the transmission of the intracellular signal upon β-adrenergic stimulation, and several intracellular substrates and transcription factors, such as PKA and CREB, having been characterized to play a role in thermogenesis.

Even though the role of the β-adrenergic receptor (β-AR) signaling pathway in BAT thermogenesis has been extensively studied, the role and directionality of insulin signaling pathway in brown adipocyte functionality, as well as the interplay between insulin and β-adrenergic signaling pathways, are less clear. Insulin has been shown to regulate adipogenesis and adipocyte differentiation, both in vivo and in vitro[15,16]. Similarly, insulin and IGF-1 receptor (IR and IGF1R) signaling is essential for the development and function of both WAT and BAT, as adipocyte-specific genetic deletion of IR and IGF1R leads to lipodystrophy both in white and brown adipose tissue[17]. Notably, in the same studies, adipose tissue-specific IGF1R ablation leads to elevated iBAT and inguinal WAT (ingWAT) Ucp1 levels, suggesting that a partial attenuation of the insulin signaling in adipose tissue may have opposing effects in its thermogenic capacity. Several studies manipulating various components of the insulin signaling pathway provide evidence towards an inverse correlation between BAT thermogenic capacity and insulin signaling pathway activity. More specifically, acute insulin injection in mice resulted in the reduction of UCP1 and PGC1-α levels[18]. Inversely, mice genetically incapable of diet-induced fasting hyperinsulinemia, generated by deletion of one of the insulin alleles, resulted in increased EE, resistance to diet-induced obesity and enhanced browning of WAT[19]. Accordingly, inducible partial gene deletion in a high-fat diet (HFD)-induced obese (DIO) mouse model, resulting in modest reduction in insulin production/secretion, leads to significant fat loss and smaller adipocytes in WAT depots[20]. Lastly, inducible adipocyte-specific deletion of IGF1R led to increased expression of Ucp1 and Pgc1-α in iBAT and ingWAT in vivo, as well as increased iBAT and reduced WAT depot weights[17]. The importance of the insulin signaling pathway and particularly of AKT in iBAT-mediated thermogenesis was additionally demonstrated by brown fat specific deletion of AKT mouse models, where it was shown that AKT2 is not required to maintain euthermia in iBAT, but its deletion enhances WAT browning[21]. Similarly, genetic deletion of AKT1 resulted in reduced weight gain upon HFD and a parallel increase in iBAT UCP1 levels[22].

AXL is a receptor tyrosine kinase (RTK), a member of the TAM family of RTKs, which also encompasses TYRO3 and MERTK. Within this family, it has been demonstrated that the AXL receptor is more potently activated by its ligand GAS6[23–25] than the other family members and intracellular signaling downstream of AXL receptor has been shown to be mediated by components of the insulin signaling pathway, such as PI3K and AKT. AXL has been extensively studied in the context of cancer as it is broadly expressed in several types of tumors such as breast cancer, non-small cell lung carcinoma, ovarian cancer, and clear cell renal carcinoma. Its expression has been linked to increased risk of metastasis and several types of tumor development in numerous studies[26–30]. Notably, GWAS studies have identified polymorphisms of AXL and GAS6 that were correlated with increased adiposity and insulin resistance[31]. Additionally, circulating GAS6 levels have been positively correlated with adiposity, insulin resistance, and inflammation in humans[32], whereas subcutaneous adipose tissue of patients with obesity demonstrated higher AXL expression, as compared to lean control subjects[33]. In earlier studies, overexpression of human AXL receptor in myeloid cells in mice, predisposed them to non-insulin dependent diabetes mellitus[34]. In line with these observations, global deletion of Axl led to a short and transient reduced weight gain[35] whereas genetic global deletion of its endogenous circulating ligand Gas6 (Gas6 -/- mouse model) resulted in reduced fat mass[36] upon a HFD challenge.

Here, we show that pharmacological and genetic inhibition of the AXL receptor protects mice against diet-induced obesity by enhancing BAT functionality, via regulating the insulin/AKT signaling pathway through the modulation of cAMP-dependent phosphodiesterase PDE3/4 activity and the transcription factors Forkhead box protein O1 (FOXO1) and ATF2.

## Results

### AXL expression is correlated with thermogenic capacity

A previously published shRNA-based kinase screening from our group demonstrated the importance of several kinases in adipocyte formation and function[37]. After filtering the positively regulated kinases for cell surface receptors and applying further criteria for their "targetability" by small-molecule kinase inhibitors and/or antagonistic antibodies, AXL receptor tyrosine kinase was identified as a promising target that could potentially regulate brown adipocyte formation. Therefore, we sought to further delineate the role of AXL receptor in brown adipose tissue and unravel putative intracellular mechanisms and systemic effects. We first evaluated the expression of the receptor in several organs and tissues in wild-type C57Bl6 mice and human samples. We could show that AXL receptor is broadly expressed in different tissues in humans[38] (Supplementary Fig. 1A). Since the adipose organ is known to exert a key role in energy homeostasis, we measured the expression of Axl in various adipose tissue depots obtained from wild-type C57Bl6 and *ob/ob* mice. We demonstrated that Axl receptor is present in all white adipose anatomical locations, but we did not observe any expression differences under obesogenic conditions (Fig. 1A and Supplementary Fig. 1B). Additionally, we sought to measure the expression of Axl in the most abundant and studied white and brown adipose depots, namely inguinal (ingWAT), epididymal (epiWAT), and interscapular brown adipose tissue (iBAT). Interestingly, we observed that Axl receptor levels are higher in both white adipose depots compared to iBAT (Fig. 1B). These results were further confirmed by bulk RNA sequencing analysis in ingWAT and iBAT depots, performed on an independent cohort of C57Bl6 male mice (Supplementary Fig. 1C). Subsequently, we wanted to investigate whether a similar expression pattern was observed in human adipose tissue samples. RNA sequencing data obtained from human subcutaneous white adipose tissue (SAT) as well as subcutaneous and deep-neck brown adipose tissue (scBAT and dnBAT, respectively), showed that AXL receptor expression was significantly higher in white compared to brown adipose tissue (Fig. 1C). The same relation was observed in brown-like and white-like adipocytes from human-derived adipocyte cell lines (hMADS) (Fig. 1D).

Axl expression pattern led us to investigate whether stimulation of thermogenic capacity of adipose depots could regulate the expression of Axl receptor. Therefore, we stimulated immortalized brown adipocyte mouse cells (iBAs)[39], which robustly express Axl receptor (Supplementary Fig. 1D) but not the other two members of the TAM family of receptor tyrosine kinases (RTKs) (Supplementary Fig. 1E–G), using isoproterenol. We could show that isoproterenol

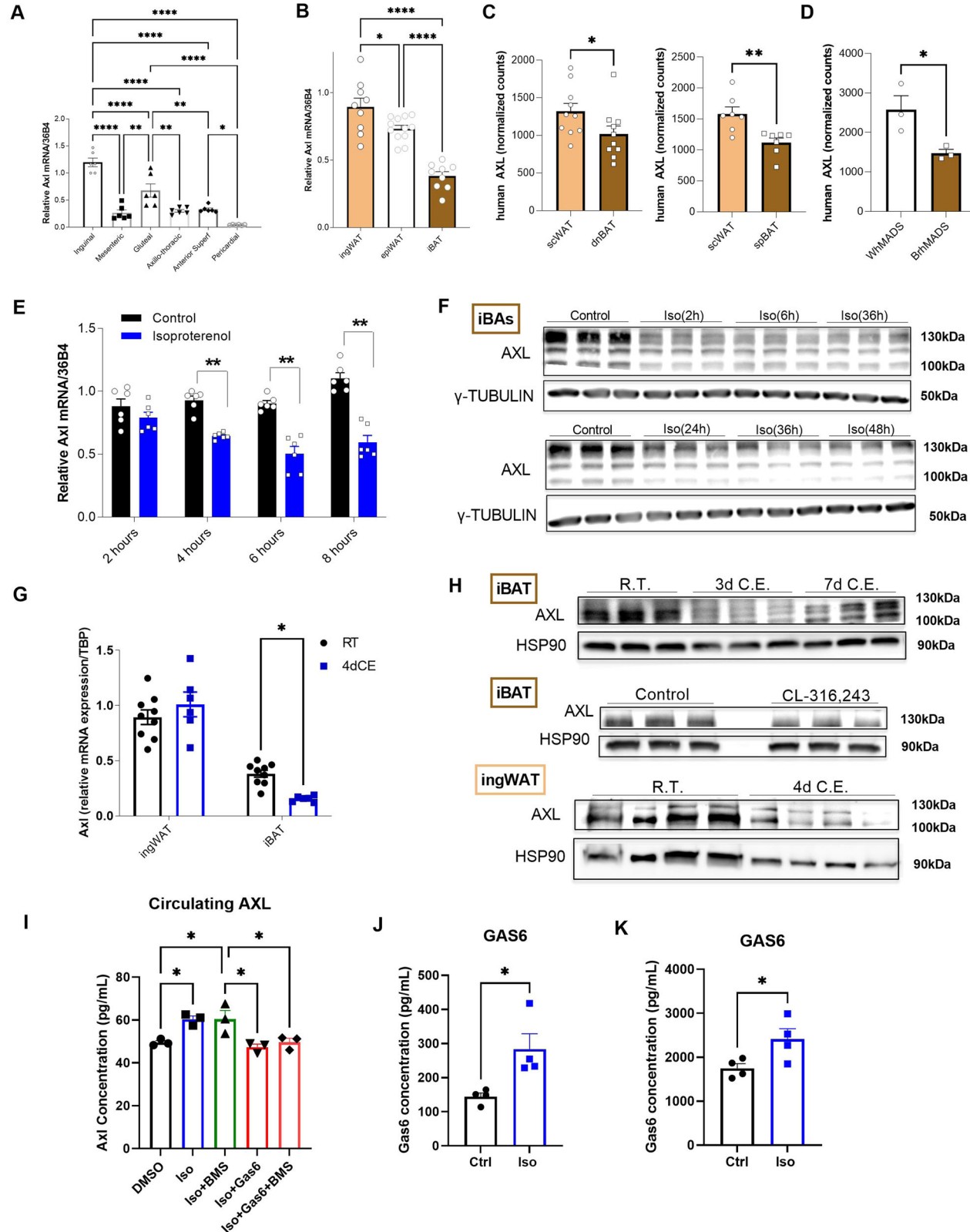

stimulation caused a significant reduction in Axl receptor mRNA expression after only 4 hours of incubation (Fig. 1E). Interestingly, reduction of AXL receptor protein levels upon isoproterenol stimulation preceded its mRNA downregulation, as its protein levels were reduced after 2 hours of β-adrenergic induction until up to 48 hours post-stimulation, both in mature iBAs and in undifferentiated pre-iBAs (Fig. 1F).

Subsequently, we sought to examine the effect of β-adrenergic stimulation on AXL receptor expression, in vivo. Therefore, we stimulated β-adrenergic signaling in wild-type mice, either by exposing them to cold temperature (8 °C) (CE) or by injecting them with the selective β-adrenergic agonist CL-316-243 (CL). We observed that 4 days of CE resulted in a significant reduction in Axl receptor mRNA levels in the iBAT depot (Fig. 1G), while Axl receptor mRNA expression

**Fig. 1 | Expression of AXL receptor is inversely regulated with thermogenic capacity. A** mRNA expression of Axl receptor in WAT depots of wild-type (WT) C57BL/6 mice ($n = 6$/group). **B** mRNA expression of Axl receptor in ingWAT, epiWAT and iBAT from WT mice (ingWAT $n = 9$; epiWAT $n = 12$; iBAT $n = 9$). **C** RNA sequencing normalized counts of AXL receptor expression levels in subcutaneous WAT (scWAT) vs. deep-neck BAT (dnBAT) (left panel – $n = 10$/group) and in subcutaneous WAT (scWAT) vs. subplatysmal BAT (spBAT) (right panel – $n = 7$/group) from human subjects. **D** RNA sequencing results (in RPKM) of AXL receptor expression levels in brown (BrhMADS) vs. white (WhMADS) cultured hMADS ($n = 3$/group). **E** mRNA expression of Axl receptor in mature iBAs after 2, 4, 6, and 8 hours of isoproterenol (2 μM) stimulation ($n = 6$/group). **F** Representative western blots of AXL receptor expression time-course of isoproterenol (Iso) stimulation from 2 to 48 hours, in iBAs; γ-TUBULIN was used as loading control. **G)** mRNA expression of Axl receptor in ingWAT and iBAT of WT mice after 4 days (4d) of cold exposure (CE)

or room-temperature (RT) ($n = 10$/group) **H)** Representative western blots of AXL receptor in iBAT or ingWAT of CE or RT acclimatized mice and in iBAT of CL-316,243 (1uM) vs. saline (Control) injected mice. HSP90 or γ-TUBULIN were used as loading controls. **I** ELISA measurement of soluble AXL in the medium of mature iBAs (after 8 days of differentiation) in response to isoproterenol (Iso), the pharmacological AXL inhibitor BMS-777607 (BMS), and/or the AXL agonist GAS6 (8 h stimulation) ($n = 3$/group). **J** ELISA measurement of GAS6 in the supernatant medium of mature iBAs after 6 h of isoproterenol treatment ($n = 4$/group). **K** ELISA measurement of GAS6 in plasma of wild-type C57/Bl6 mice after 16 h of treatment with CL-316,243 (1uM) or saline (Control) ($n = 4$/group). For all graphs, results are presented as average ± SEM *$p < 0.05$, **$p < 0.01$, ***$p < 0.001$ and ****$p < 0.0001$ as compared to respective controls. For two group comparisons (**C**–**E**, **G**, **J**, **K**) unpaired two-tailed t-test was performed, for three or more group comparisons (**A**, **B**, **I**) one-way ANOVA was performed. Tukey test was applied to correct for multiple comparisons.

was unchanged upon CE in ingWAT (Fig. 1G). AXL receptor protein levels were significantly reduced upon both exposure to cold for 2-3 days as well as by CL administration (Fig. 1H). The extracellular domain of AXL receptor is known to be cleaved, which releases soluble AXL into circulation[40]. We observed that isoproterenol stimulation of iBAs resulted in increased levels of soluble AXL in the medium, an effect, which was maintained in the presence of a pharmacological AXL inhibitor (BMS-777607). Co-incubation with the AXL receptor agonist GAS6 blunted the isoproterenol-stimulated induction of soluble AXL in the medium of brown adipocytes. Taken together, these results point towards a post-transcriptional and post-translational regulation of AXL receptor in response to β-adrenergic receptor activation. Furthermore, we observed that GAS6 is upregulated by ~2-fold in the supernatant of mature iBAs in response to isoproterenol treatment (Fig. 1J). Similarly, circulating GAS6 is significantly upregulated in the plasma of mice after injection of the β-adrenergic receptor agonist CL-316,243 (Fig. 1K).

## Pharmacological inhibition of AXL receptor enhances brown adipocyte functionality

The inverse correlation between AXL expression and thermogenic capacity of adipose depots as well as the regulation of its expression in response to β-adrenergic stimulation suggested a potential role of AXL receptor in adipose tissue function and led us to investigate whether manipulation of its activity could regulate the functionality of brown and white adipocytes. Therefore, we employed three commercially available small molecules that function as pharmacological AXL receptor inhibitors: NPS-1034 (NPS)[41], BMS-777607 (BMS)[42,43] and R428[44]. Binding affinities of these three compounds for the other members of the TAM (AXL, TYRO3, MERTK) or MET (MET and RON) family of proteins are shown in Supplementary Table 1. The use of the above-mentioned inhibitors is enabled by the fact that the other TAM proteins are almost undetectable in iBAs (Supplementary Fig. 1D–F). Potencies of the three compounds for AXL receptor are as follows: NPS-1034 with IC50 of 10.3 nM, BMS-777607 with IC50 of 1.1 nM and R428 with IC50 of 14 nM. To test their effect, we differentiated immortalized mouse-derived brown adipogenic progenitors (pre-iBAs) for 5 days – period during which cells are differentiated into mature brown adipocytes – and subsequently (during days 5-8 of terminal differentiation) treated them with increasing dosages of the three different pharmacological AXL receptor inhibitors. Treatment with these compounds increased levels of Ucp1, an effect that was observed in both isoproterenol-stimulated (Fig. 2A) and unstimulated brown adipocytes (Fig. 2B) in a dose-dependent manner. Additionally, we observed that pharmacological inhibition of AXL receptor resulted in the upregulation of several other BAT-enriched genes such as the lipid-droplet associated genes Elovl3 and Cidea and the mitochondrial genes Cox7a1 and Cox8b (Supplementary Fig. 2A). Similarly, transcription factors known to be involved in the acquisition of the brown adipocyte phenotype and in mitochondrial biogenesis, such as Pgc1-α,

C/ebp-α, and C/ebp-β were also significantly upregulated upon treatment with the pharmacological inhibitors (Fig. 2A, B and Supplementary Fig. 2A).

High-throughput immunofluorescence analysis and western-blot measurements revealed that treatment of mature brown adipocytes with pharmacological AXL receptor inhibitors similarly led to a dose-dependent increase in UCP1 levels (Fig. 2C and D). This effect was not dependent on changes in differentiation or proliferation of adipocytes (Supplementary Fig. 2B) but can rather be attributed to a total increase in the number of UCP1 positive cells (Fig. 2C) as well as to a total increase in the amount of UCP1 protein per adipocyte (Supplementary Fig. 2B). Similarly, PGC1α protein levels were elevated upon pharmacological AXL receptor inhibition with NPS and BMS (Supplementary Fig. 2C). Subsequently, we sought to evaluate whether pharmacological inhibition of AXL receptor could change the metabolic phenotype and functionality of white adipocytes. Therefore, we differentiated immortalized mouse-derived white adipogenic progenitors (pre-iWAs)[45] for 5 days – the period during which cells are differentiated into mature white adipocytes – and subsequently (during days 5-8 of terminal differentiation) treated them with increasing dosages of the two different pharmacological AXL receptor inhibitors. Pharmacological AXL receptor inhibition resulted in a significant upregulation of Ucp1 and Pgc1-α as well as mitochondrial markers such as Cox8b, indicative of an enhanced brown-like oxidative phenotype in these cells (Supplementary Fig. 2D).

In order to investigate the human relevance of the previously described in vitro findings and confirm their validity and reproducibility across several cell-culture models, we tested the effect of the pharmacological inhibitors BMS and R428 in human subcutaneous-derived immortalized adipocytes (hMADS)[46]. Utilizing a similar setup, we differentiated hMADS[46] for 14 days (the period during which cells are differentiated into mature brown adipocytes) and subsequently (during days 15-21 of terminal differentiation) treated them with increasing dosages of the two different pharmacological AXL receptor inhibitors. Similar to the effects that we observed on the immortalized mouse cell lines, treatment of differentiated mature brown hMADS with the AXL receptor inhibitor compounds led to a significant upregulation of the thermogenic brown-adipocyte specific thermogenic gene UCP1 and enhanced the levels of the mitochondrial biogenesis-associated transcription factor PGC1-α (Supplementary Fig. 2E).

Next, we hypothesized that activating AXL receptor would exert the opposite effect, i.e., suppress the expression of thermogenic genes in iBAs. To test this, we used an agonistic AXL receptor antibody, in the assay described above, treating mature iBAs during days 5-8 of differentiation with increasing concentration of this AXL receptor signaling-activating antibody. Despite a mild trend towards suppression of gene expression, we did not observe any significant downregulation of Ucp1 (Supplementary Fig. 2F), Pgc1-α (Supplementary Fig. 2G) or Cox8b (Supplementary Fig. 2H) in mature iBAs in response to activation of AXL receptor, suggesting that stimulation above basal

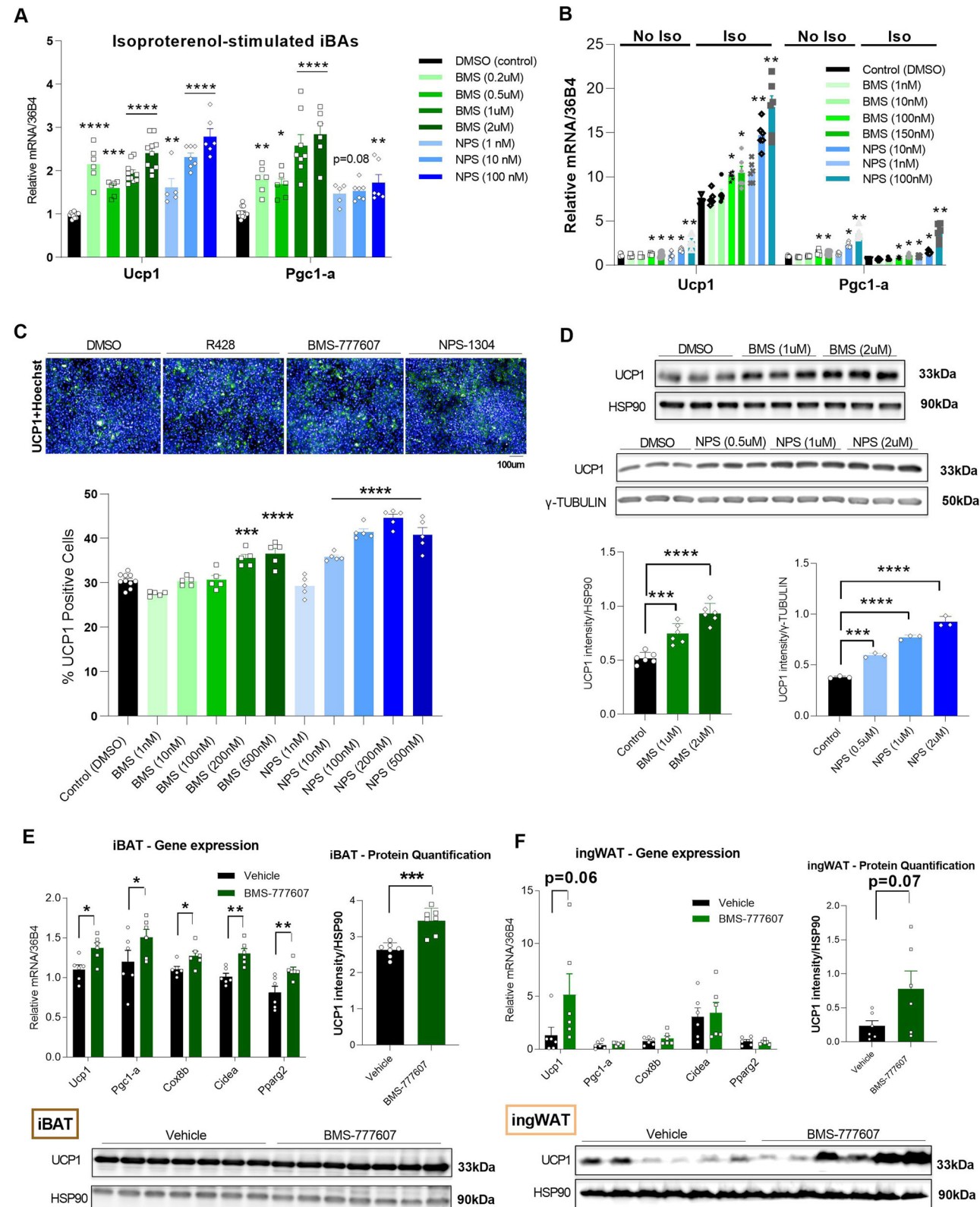

activation of the AXL receptor does not exert the opposite phenotypic effects as compared to its inhibition.

Having characterized the effects of pharmacological inhibition of AXL receptor in vitro, we investigated whether similar effects could be recapitulated, in vivo. Therefore, we orally administered BMS or vehicle control for 7 consecutive days, by gavage in C57Bl6 mice that had been fed a HFD for 12 weeks. BMS administration led to an

induction of Ucp1 (protein and mRNA level) and other thermogenic and mitochondrial genes in iBAT (Fig. 2E), as well as a robust increase of Ucp1 (protein and mRNA) in ingWAT (Fig. 2F) but not in the epiWAT (Supplementary Fig. 2I). These observations were reproduced in a second cohort in which BMS was administered for two consecutive weeks as dietary admixture, as demonstrated by elevated levels of Ucp1 (Supplementary Fig. 2J), coupled to an increase in energy

**Fig. 2 | Pharmacological inhibition of Axl receptor enhances brown adipocyte functionality. A–D** Mature iBAs were treated with AXL inhibitors BMS-777607 (BMS, green), NPS-1304 (NPS, blue), or DMSO (black). On day 8 of differentiation, iBAs were either harvested unstimulated or were stimulated with isoproterenol (1 µM) for 6-8 hours. A-B) qRT-PCR analysis. Representative graph of 4-6 independent experiments showing mRNA expression of Ucp1 and Pgc1-α in (**A**) isoproterenol-stimulated (Control-BMS1uM $n = 12$; BMS0.2uM-0.5uM-NPS100nM $n = 6$; BMS2uM $n = 10$; NPS1nM $n = 5$; NPS10nM $n = 7$) and B) unstimulated and isoproterenol-stimulated mature iBAs after treatment with increasing concentrations of AXL inhibitors ($n = 6$/group). **C** Percentage of UCP1-positive isoproterenol-stimulated mature iBAs after treatment with AXL inhibitors. Representative immuno-fluorescent images (A488/green: FoxO1, blue/A497: Hoechst/Nuclei) (upper) and quantitative analysis (lower) of 3-4 independent experiments of high-throughput immunofluorescent imaging analysis (Control $n = 10$; all other groups $n = 5$/group). **D** Representative western blots and respective quantifications of UCP1 in iBAs after treatment with AXL inhibitors. **E**, **F** Chronic oral administration of BMS-777607 in C57Bl6 DIO mice. Graphs show gene expression and protein expression analysis of thermogenesis-associated genes in iBAT (**E**) and ingWAT (**F**) of mice treated with the compound or vehicle for 14 consecutive days ($n = 6$/group). For all graphs, results are presented as average ± SEM. *$p < 0.05$, **$p < 0.01$, ***$p < 0.001$, ****$p < 0.0001$ as compared to respective controls. For two group comparisons (**E, F**) unpaired two-tailed t-test was performed, for three or more group comparisons (**A**–**C**, **F**) multiple comparisons one-way ANOVA was performed (Tukey test was applied to correct for multiple comparisons).

expenditure (Supplementary Fig. 2K). Taken together, we show here that pharmacological inhibition of AXL receptor leads to enhanced brown adipocyte activity and can induce uncoupled thermogenesis, both in vitro and in vivo.

## AXL receptor knockdown enhances brown adipocyte functionality

To corroborate the pharmacological data, we used genetic approaches to regulate the expression and activity of AXL receptor. Therefore, we performed siRNA-mediated knockdown of Axl by transfecting mature adipocytes on day 5 of their differentiation and measuring gene expression and other parameters on day 8 of their terminal differentiation. First, we demonstrated that siRNA-mediated knockdown of Axl receptor efficiently reduced the levels of the receptor by more than 90%, both at the mRNA and at the protein level, whereas the expression of other TAM receptors was unchanged (Fig. 3A and B). Subsequently, we evaluated the effect of the siRNA-mediated deletion of Axl receptor on the thermogenic capacity of mature iBAs. We could demonstrate that Axl receptor knockdown in mature brown adipocytes caused a significant upregulation of Ucp1, Elovl3, and Cidea as well as mitochondrial genes such as Cox7a1 and Cox8b, and transcriptional regulators such as Pgc1-α, Pparγ, and C/ebp-β. Similar to the pharmacological experiments, this enhanced brown adipocyte phenotype was observed in both unstimulated and isoproterenol-stimulated mature brown adipocytes (Fig. 3C). As expected, siRNA-mediated knockdown of Axl receptor significantly upregulated UCP1 protein levels both in unstimulated and isoproterenol-stimulated mature iBAs (Fig. 3D).

The enhanced thermogenic machinery of brown adipocytes in response to Axl receptor knockdown, led us to investigate whether these transcriptional and protein cellular changes could be translated into functional changes. Therefore, we measured oxygen consumption rate (OCR) in mature iBAs, 2-3 days after Axl receptor knockdown. In accordance with the increased expression of thermogenic genes, we could show that siRNA-mediated Axl receptor deletion enhanced isoproterenol-stimulated mitochondrial respiration, isoproterenol-stimulated uncoupled respiration as well as maximal respiration (Fig. 3E, Supplementary Fig. 3A–C).

To investigate the translation of these findings to a human model, we performed the siRNA-mediated AXL receptor knockdown experiments on mature brown-like hMADS, during days 14-18 of their differentiation and quantified the transcriptome of these cells. Transcriptomic analysis showed numerous differentially expressed genes between the AXL receptor-knockdown and the non-targeted control adipocytes (Supplementary Fig. 3D). GO enrichment analysis based on the differentially expressed genes revealed a significant enrichment for lipid metabolic processes (Fig. 3F), whereas subsequent KEGG pathway analysis demonstrated that the FOXO, PI3K/AKT, PPAR, and RAS signaling pathways were significantly upregulated in AXL receptor-knockdown brown-like hMADS, compared to non-targeted siRNA controls (Fig. 3G, Supplementary Fig. 3E). These transcriptomic changes were translated into enhanced cAMP-stimulated mitochondrial and cAMP-stimulated uncoupled respiration as well as in maximal respiration in hMADS upon AXL receptor knockdown, as compared to the non-targeted controls (Fig. 3H and Supplementary Fig. 3F). Taken together, these data demonstrate that ablation of AXL receptor expression leads to increased brown adipocyte development as well as increased mitochondrial respiration.

## AXL receptor inhibition induces FOXO1 nuclear localization by modulating the Insulin/PI3K/AKT axis

Next, we sought to investigate the potential molecular downstream mechanisms via which inhibition of AXL receptor activity may lead to the enhanced thermogenic brown-like phenotype of brown and white adipocytes. Based on pathway enrichment analysis we performed in hMADS, we could show that PI3K/AKT and FOXO signaling pathways were significantly regulated in response to AXL receptor knockdown. Therefore, we focused on analysing the regulation of the insulin receptor (IR)/AKT axis. Initially, we investigated whether there was any interference between the Insulin/IR and GAS6/AXL pathways by measuring AKT activity in cultured immortalized brown adipocytes upon activation and/or inhibition of AXL receptor activity. Therefore, we acutely treated mature iBAs with the AXL receptor agonist GAS6 or with the AXL receptor inhibitor BMS or with a combination of both GAS6 and BMS simultaneously, in the presence or absence of 10 nM insulin. As expected, we show that GAS6 induced AXL receptor phosphorylation whereas BMS inhibits its phosphorylation, either in the presence or absence of GAS6 in mature iBAs (estimated IC50 of 0.5 µM for BMS-777607; Fig. 4A). Basal levels of pAXL are observed (DMSO control) and BMS reduces pAXL, even in the absence of GAS6 stimulation, suggesting basal constitutive activation of AXL receptor (Fig. 4A). Additionally, using an activating AXL receptor-specific antibody, we measured the phosphorylation levels of AXL receptor and AKT. We demonstrate a robust induction of pAXL and pAKT in response to the AXL-activating antibody, thus confirming the AXL receptor - AKT signaling axis in mature brown adipocytes, independent of protein expression changes of AXL receptor and its ligand GAS6 (Fig. 4B).

Subsequently, we measured acute phosphorylation levels of AKT and could show that both unstimulated and insulin-stimulated AKT phosphorylation on T308 and S473 residues were attenuated upon pharmacological inhibition of AXL receptor whereas they were mildly enhanced upon GAS6 activation (Fig. 4C). Notably, this effect was observed both under low (10 nM) and high (100 nM) insulin concentrations, and the same findings could be reproduced in other cell lines, such as the brown-like hMADS (Supplementary Fig. 4A). Pharmacological AXL receptor inhibition did not affect the phosphorylation of IR, IRS-1, PDGFR-α or PDGFR-β (Supplementary Fig. 4B). Downstream phosphorylation levels of STAT3 and STAT5 were not significantly affected either, in response to AXL receptor inhibition (Supplementary Fig. 4H). As AXL receptor appears to modulate insulin-stimulated activity of AKT, we measured the levels of phosphorylation

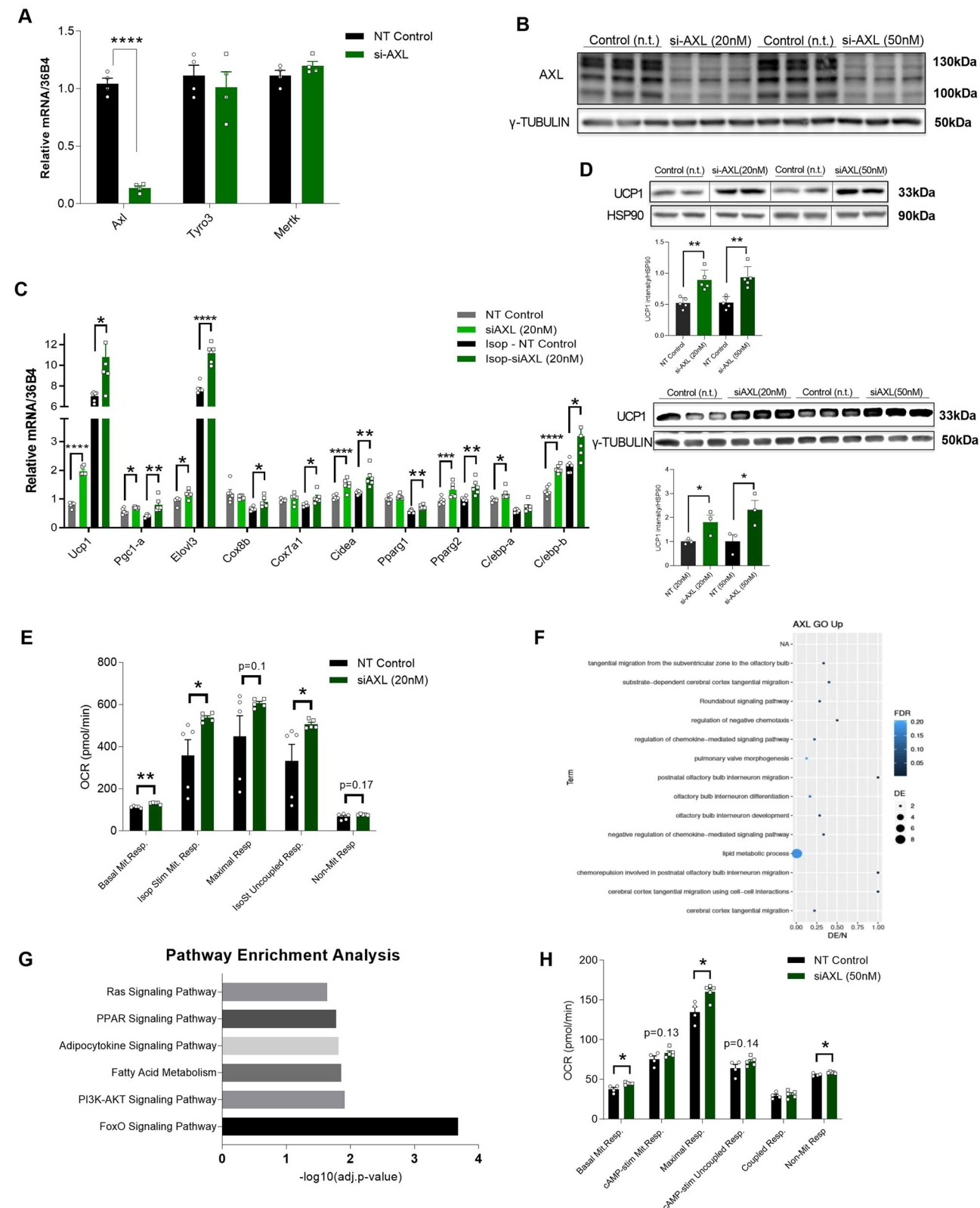

of AKT upon AXL receptor inhibition in a time-dependent manner both in the presence of low and high doses of insulin. Pharmacological inhibitor BMS significantly attenuated the insulin-stimulated AKT phosphorylation at all measured time points within 5 min post-insulin stimulation and persisted for more than 2 hours (Fig. 4D). Interestingly, we noticed that despite reduced phosphorylation of AKT in both low and high insulin concentrations, the AXL receptor

inhibitor-induced attenuated pAKT levels upon high insulin dosage (100 nM) stimulation still surpasses pAKT levels of the low-dosage (10 nM) insulin-stimulated AXL inhibitor-untreated cells (Fig. 4D). In other words, pharmacological AXL receptor inhibition moderately attenuates the activity of the insulin/AKT signaling pathway, whereas the state major regulator is insulin signalling. Taken together, we show that regulation of the activity of AXL receptor modulates the

**Fig. 3 | siRNA-mediated knockdown of Axl receptor enhances brown adipocyte functionality. A–F** Mature iBAs (day 4-5 of differentiation) were treated with siRNA targeting Axl receptor. 48–72 hours post-siRNA transfection, cells were either unstimulated or stimulated with isoproterenol (1 μM) for 6-8 hours. **A** Gene expression analysis of Axl, Tyro3, and Mertk (n = 4/group). **B**) Representative western blot of at least 3 independent experiments, of AXL receptor upon siRNA-mediated knockdown in mature iBAs. γ-TUBULIN was used as loading control. **C** Representative graph of 3 independent gene expression analyses of thermogenesis- and adipogenesis-associated genes in response to siRNA-mediated knockdown of Axl receptor in unstimulated or isoproterenol-stimulated mature iBAs (n = 6/group). **D** Representative western blots of 3 independent experiments and respective quantifications of UCP1 in response to siRNA-mediated knockdown of AXL receptor in unstimulated (upper) or isoproterenol-stimulated (lower) mature iBAs. **E** Calculations of basal, isoproterenol-stimulated, maximal,

uncoupled, and non-mitochondrial respiration in mature iBAs upon siRNA-mediated knockdown of Axl receptor based on oxygen consumption rate (OCR – Seahorse assays) measurements (n = 5/group). **F–H** Brown hMADS were differentiated into mature brown adipocytes and treated with siRNA against AXL receptor (Day 14). 4 days post-siRNA transfection, cells were either harvested or used for functional assays. **F, G** Gene expression via bulk RNA sequencing analysis. GO (**F**) and KEGG (**G**) pathway enrichment analyses in response to siRNA-mediated knockdown of AXL receptor in brown hMADS. **H** Calculations of basal, cAMP-stimulated, maximal, uncoupled, coupled, and non-mitochondrial respiration in brown hMADS upon siRNA-mediated knockdown of AXL receptor based on OCR measurements (n = 5/group). For all graphs, results are presented as average ± SEM. *$p < 0.05$, ***$p < 0.001$ as compared to respective controls. Unpaired two-tailed t-test was performed for all comparisons, statistically significant differences are annotated on the graph.

phosphorylation of AKT by slightly attenuating and finely tuning the insulin-mediated AKT signaling pathway.

As we previously demonstrated, pathway analysis of the differentially expressed genes upon siRNA-mediated AXL receptor knockdown pointed toward a significant regulation of the FOXO signaling pathways. Hence, we measured the phosphorylation of FOXO1 in response to inhibition of AXL receptor in insulin-stimulated mature brown adipocytes and we could show that pharmacological inhibition of AXL receptor led to reduced insulin-stimulated phosphorylation of FOXO1 (Supplementary Fig. 4B). The reduced FOXO1 phosphorylation occurs within 2 min and last for up to 60 min after insulin stimulation (Supplementary Fig. 4C). Taking into consideration the well-established effect of FOXO1 phosphorylation on its nuclear-to-cytoplasmic translocation, we sought to evaluate FOXO1 cytoplasmic and nuclear localization in response to AXL signaling regulation. First, we maximally stimulated mature iBAs using a high concentration of insulin (20 nM) in the presence of FBS, which contains multiple growth factors, as a positive control condition of an almost complete nuclear exclusion and cytoplasmic translocation of FOXO1 (Fig. 4E, left panel). Subsequently, we stimulated mature iBAs with a lower dosage of insulin (10 nM), in the presence of the pharmacological AXL receptor inhibitor BMS, the AXL receptor agonist GAS6, or DMSO control. We observed that AXL receptor inhibition prevented the low-dosage insulin-stimulated FOXO1 translocation from the nucleus to the cytoplasm as compared to DMSO-treated control cells, whereas in the presence of GAS6 the majority nuclei appeared to be FOXO1 negative. Consistent with a reduced phosphorylation of pAXL and pAKT in response to co-incubation with BMS and GAS6, the majority of nuclei of iBAs that were simultaneously treated with BMS and GAS6, stained positive for FOXO1 (Fig. 4E). We observed the same effects in differentiated cultured mouse adipose-derived mature white adipocytes (Supplementary Fig. 4D). Thus, insulin-stimulated brown and white adipocytes maintain prolonged and enhanced nuclear FOXO1 localization in response to pharmacological inhibition of AXL receptor (Fig. 4E and Supplementary Fig. 4D). Remarkably, the effect of enhanced nuclear FOXO1 localization in response to AXL receptor inhibition was even more striking under milder stimulation conditions, when using simply FBS-containing medium (Fig. 4F). Contrary, in FBS-depleted non-insulin stimulated brown adipocytes, FOXO1 was almost exclusively localized in the nucleus, both in the presence or absence of AXL receptor inhibitors. Interestingly, the previously observed GAS6-stimulated increase in AKT phosphorylation (Fig. 4C) was not sufficient to cause a significant GAS6-stimulated nuclear exclusion of FOXO1 translocation, but it led to a rather mildly enhanced FOXO1 translocation to the cytoplasm (Fig. 4E – right panel), possibly due to the basal AXL activation from endogenous GAS6. As expected, pharmacological inhibitors of AKT recapitulated the phenotype of AXL receptor inhibitors and led to a blunted insulin-stimulated AKT phosphorylation (Supplementary Fig. 4F) and a corresponding prolonged enhanced FOXO1 nuclear localization in a dose-dependent manner (Supplementary Fig. 4G). Subsequently, we asked

whether the same effect can be observed under supra-physiological conditions of insulin stimulation. Therefore, we repeated the acute-stimulation experiments in the presence of a high dose of insulin (100 nM). Remarkably, we did not observe the same effect of enhanced FOXO1 nuclear localization in response to AXL inhibition compared to the previously described low-grade insulin (10 nM) conditions, as FOXO1 appeared almost entirely localized in the cytoplasm (Supplementary Fig. 4G). Taken together, our data suggest that a threshold - likely as a continuum and less as an "on-off switch" - exists for AKT phosphorylation due to a balance between the activities of the Gas6/AXL receptor and insulin/IR (or other growth factors, e.g., IGF1R) pathways, the relative activity of which ultimately determines the subcellular localization of FOXO1.

To identify whether FOXO1 mediates the response of the previously observed AXL-inhibition-enhanced gene expression of Ucp1 and Pgc1-α in iBAs under non-isoproterenol-stimulated conditions (Fig. 2A), we co-incubated mature brown adipocytes (during days 6-8 of "terminal" differentiation) with both the AXL receptor inhibitor BMS and a pharmacological FOXO1 inhibitor. As expected, AXL receptor inhibition (BMS incubation) led to a significant increase in gene expression of Ucp1 and Pgc1-α. Strikingly, pharmacological FOXO1 inhibition dramatically diminished Ucp1 and Pgc1-α gene expression and blunted the BMS-induced stimulation of these genes, suggesting that FOXO1 is required for the expression of these genes in iBAs and that it mediates the AXL-inhibition-stimulated phenotype (Fig. 4G). To corroborate our pharmacological approach, we performed siRNA-mediated knockdown of FoxO1 in mature brown adipocytes in the presence or absence of BMS. We observed a significant reduction in Ucp1 and Pgc1-α gene expression in response to FoxO1 knockdown and an impaired response of iBAs to the pharmacological AXL inhibition (Fig. 4H). Overall, these data suggest that FOXO1 is a crucial transcriptional regulator of the expression of Ucp1 and Pgc1-α in iBAs and it mediates the effects of pharmacological AXL inhibition in enhanced thermogenesis.

### AXL receptor inhibition increases intracellular cAMP levels through moderate attenuation of the Insulin/AKT/PDE pathway

Leveraging the differentially expressed genes that were revealed between the AXL receptor-knockdown and control human adipocytes, we performed Enrichr analysis to detect the associated transcription factors (TFs) that may be involved in these alterations. Among the TFs that emerged as significantly enriched upon siRNA-mediated AXL knockdown in hMADS, we observed members of the cAMP-responsive CREB/ATF family of proteins, which are well-established downstream targets of the β-adrenergic/cAMP/PKA signaling pathway (Fig. 5A). Therefore, we hypothesized that pharmacological AXL inhibition could lead to increased intracellular cAMP levels via an AKT-dependent mechanism. To test this hypothesis, we acutely stimulated mature brown adipocytes with isoproterenol and a simultaneous co-incubation with AXL inhibitors and/or the AXL agonist GAS6. We

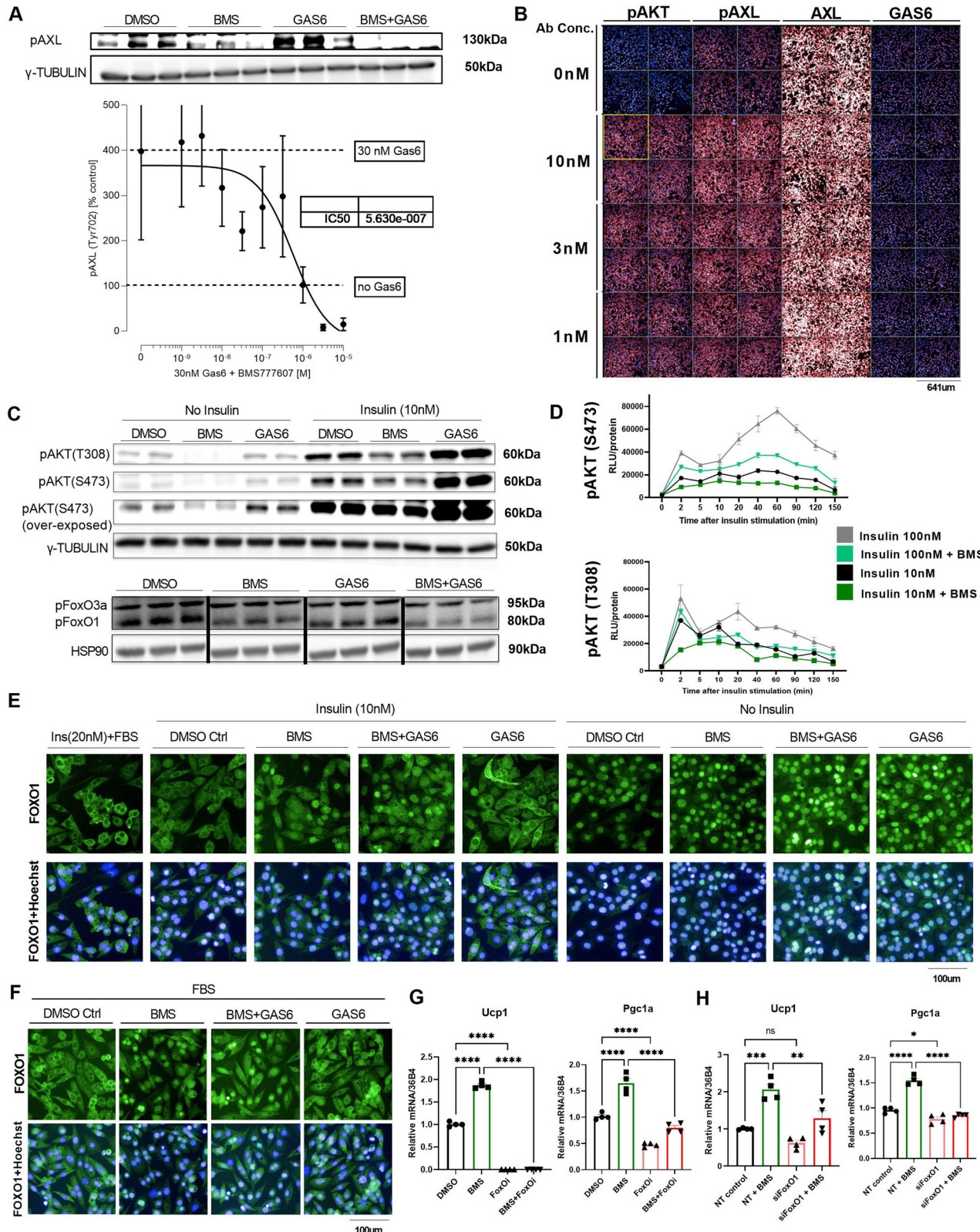

observed that isoproterenol-stimulated brown adipocytes acutely treated with pharmacological AXL inhibitors have higher cAMP levels compared to the control cells. Conversely, acute GAS6 treatment led to mildly but significantly reduced intracellular cAMP levels, in agreement with the notion of a basal constitutive activation of AXL receptor. These effects on intracellular cAMP levels were observed within 5-15 min post-isoproterenol stimulation (Fig. 5B) in

a dose-dependent manner (Supplementary Fig. 5A–C). Even the co-incubation of isoproterenol-stimulated brown adipocytes with a simultaneous treatment using both pharmacological AXL inhibitors and GAS6, resulted in increased intracellular cAMP levels (Supplementary Fig. 5D). Interestingly, these effects of AXL inhibitors were maintained at higher dosages of isoproterenol (Fig. 5C) or forskolin (Supplementary Fig. 5E). As expected, the elevated intracellular cAMP

**Fig. 4 | Pharmacological inhibition of AXL receptor induces a prolonged AKT-dependent nuclear FOXO1 localization. A** Representative western blots and quantifications (lower), of phosphorylated AXL (Tyr702) receptor in response to acute treatment with AXL inhibitor BMS-777607 and AXL agonist GAS6 (30 nM) in mature iBAs. **B** Representative immunofluorescent images of three independent experiments of pAKT, pAXL, total AXL, and GAS6 (from left to right) after treatment with AXL receptor-activating antibody, in mature iBAs. **C)** Representative western blots of three independent experiments of phosphorylated AKT (pAKT) (upper) and phosphorylated FOXO1/FOXO3a (pFOXO1/pFOXO3a) (lower) in response to treatment with BMS-777607 (10uM) and GAS6 (100 ng/ml) in unstimulated or insulin-stimulated mature iBAs. **D** Time-course effect of BMS-777607 (10uM) on AKT phosphorylation (upper: S473, lower: T308), measured by ELISA in insulin-stimulated mature iBAs at 10 nM (black and dark green lines) and 100 nM (grey and light green lines) (n = 2/group/timepoint). **E** Representative immunofluorescent images (A488/green: FOXO1, blue/A497: Hoechst/Nuclei) of three independent

experiments of FOXO1 in insulin-stimulated and non-stimulated mature iBAs after treatment with BMS and/or GAS6. Positive controls (left panel) were treated with 20 nm insulin in the presence of FBS or used unstimulated and fasted. **F** Representative immunofluorescent images (A488/green: FOXO1, blue/A497: Hoechst/Nuclei) of three independent experiments of FOXO1 in FBS-stimulated mature iBAs in response to BMS-777607 and/or GAS6. **G** Gene expression analysis of Ucp1 and Pgc1-α in mature iBAs in response to pharmacological inhibition of FOXO1 with AS1842856, in the presence or absence of BMS-777607 (n = 4/group). **H** Gene expression analysis of Ucp1 and Pgc1-α in mature iBAs in response to siRNA-mediated knockdown of FoxO1, in the presence or absence of BMS-777607 (n = 4/group). For all graphs, results are presented as average ± SEM (n = 3-6). *p < 0.05, **p < 0.01, ***p < 0.001. For three or more group comparisons (**G, H**), one-way ANOVA was performed (Tukey test was applied to correct for multiple comparisons). Statistically significant differences are annotated in the graph.

levels in response to AXL inhibition, led to an enhanced phosphorylation of PKA substrates (Fig. 5D), such as CREB and ATF (Fig. 5E) and translated into stimulation of brown adipocyte functionality, as evaluated by an acute increase in the oxygen consumption rate (Fig. 5F and Supplementary Fig. 5F).

To examine whether AXL receptor inhibition-induced elevated intracellular cAMP levels can be attributed to a crosstalk between insulin/IR and GAS6/AXL receptor signaling pathways, we used pharmacological inhibitors and activators of key established downstream substrates of these pathways and measured their relative contribution to intracellular cAMP levels. First, we utilized a specific AKT activator and acutely treated isoproterenol-stimulated mature adipocytes, in the presence and/or absence of AXL inhibitor BMS. We observed that AKT activation mitigated the effect of AXL inhibition on cAMP levels, as combined AXL inhibition and AKT activation did not lead to elevated cAMP levels, as compared to controls in response to isoproterenol (Fig. 5G – left panel). Contrary, pharmacological AKT inhibition recapitulated the phenotype of AXL inhibition, resulting in increased cAMP levels, whereas combined AXL and AKT inhibitions did not lead to an additive effect (Fig. 5G – right panel). Next, we used a specific PI3K inhibitor, which recapitulated the phenotypes of AXL and AKT inhibitors. In fact, PI3K inhibition led to increased cAMP levels, whereas its combination with AXL inhibition resulted in a small additive effect (Fig. 5H – left panel). Contrary, PTEN inhibition (Supplementary Fig. 5J) blunted the AXL-inhibition-induced cAMP effect, as co-incubation of isoproterenol-stimulated brown adipocytes with both AXL receptor and PTEN inhibitors enhanced the activity of the insulin signaling pathway (pAKT) and led to a reduction of cAMP levels (Fig. 5H – right panel).

To further corroborate that AXL receptor inhibition enhanced brown adipocyte functionality via attenuated AKT phosphorylation, we employed a previously characterized anti-AXL receptor monoclonal antibody[47]. We observed that despite the antibody's capacity to reduce total protein levels of AXL receptor in A549 cells (Fig. 5I), it agonizes AXL receptor activity as evidenced by increased pAKT levels in mouse iBAs (Fig. 5J). Based on these data, we investigated the effect of this agonistic-depleting AXL receptor antibody on energy expenditure in mature brown adipocytes. Acute co-administration of isoproterenol-stimulated brown fat cells with the agonistic AXL receptor antibody led to a significantly attenuated oxygen consumption rate (OCR) (Fig. 5K). Accordingly, AXL receptor antibody stimulation led to reduced intracellular cAMP levels in isoproterenol-stimulated brown adipocytes (Supplementary Fig 5G). Subsequently, we sought to investigate which specific cellular substrate could potentially mediate the observed regulation in the intracellular cAMP levels upon AXL receptor inhibition or activation, as AKT is not known to directly interact with cAMP. Therefore, we focused on the function of the PDEs, which are known regulators of the intracellular cAMP levels[48]. To obtain more conclusive mechanistic data on the role of

individual PDEs, we used the agonistic AXL antibody in an established commercial assay that measures PDE activity separately for each phosphodiesterase isoform. We observed that the agonistic anti-AXL antibody given acutely stimulated the activity of PDE3 and PDE4 (Fig. 5L). Based on these observations, we additionally used pharmacological compounds to specifically inhibit the activity of PDE3 and PDE4, (PDE3i and PDE4i, respectively) hypothesizing that their inhibition will exert an opposite effect as compared to the agonistic anti-AXL antibody and will recapitulate the phenotype of pharmacological AXL receptor inhibitors. Indeed, acute pharmacological inhibition of PDE3 (Supplementary Fig. 5H) or PDE4 (Supplementary Fig. 5I) resulted in elevated intracellular cAMP levels in isoproterenol-stimulated brown adipocytes, with PDE4i showing a stronger effect; suggesting that PDE inhibition mediates the effect of AXL receptor inhibition. Collectively, these data demonstrate a role of the AXL/AKT/PDE axis in regulating intracellular cAMP levels and energy expenditure in brown adipocytes.

Lastly, we sought to test whether these identified signaling pathways and TFs mediate not only the AXL-inhibition-induced cAMP response, but also the resulting thermogenesis-associated gene expression in mature brown adipocytes. First, we tested whether GAS6 induces phosphatidyl-serine (PtdSer), which has been previously described to occur in response to activation of members of the TAM family[49]. We observed no effect of GAS6 on PtdSer levels (Supplementary Fig. 5K). Next, we focused on the canonical cAMP/PKA pathway and downstream targets. Therefore, we treated mature brown adipocytes during days 6-8 of their terminal differentiation with the well-established pharmacological PKA inhibitor compound H89, in the presence and/or absence of AXL inhibitors. Strikingly, PKA inhibition entirely abolished expression of Ucp1 and Pgc1a and blunted the effect of AXL receptor inhibition, suggesting that the cAMP/PKA pathway mediates the AXL inhibition phenotype (Fig. 5M). Taking into consideration the TF enrichment analysis (Fig. 5A), we additionally hypothesized that the transcription factor ATF2 mediates the effect of AXL receptor inhibition on gene expression. Therefore, we knocked down Atf2 in iBAs, in the presence and/or absence of AXL inhibitors. Notably, Atf2 knockdown entirely abolished the upregulation of AXL inhibition on enhanced gene expression of Ucp1 and Pgc1-α, suggesting that it mediates its effect (Fig. 5N).

## Constitutive global and inducible adipocyte-specific deletion of AXL receptor enhances the thermogenic signature of adipose tissues and protects from diet-induced obesity

We next sought to investigate the effects of genetic manipulation and deletion of AXL receptor in vivo. Therefore, we generated the constitutive Axl receptor global (whole-body) knockout mouse model (*Axl KO*) (Fig. 6A). First, we characterized the phenotype of these mice under regular (room-temperature and chow diet) housing conditions. We observed that *Axl KO*, as well as the *Axl* heterozygous (*Het*), mice did not show any differences in body weight upon birth as compared

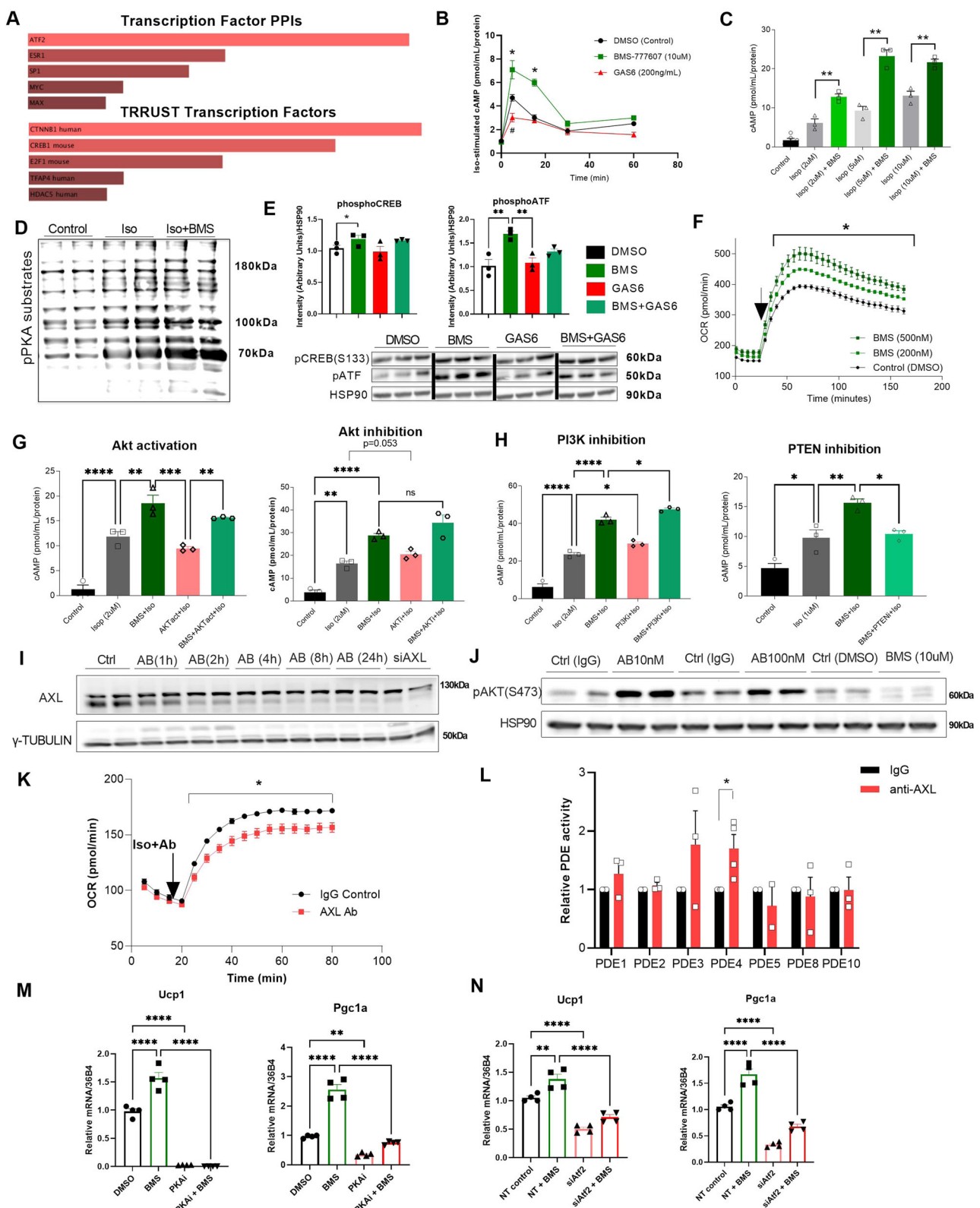

to their wild-type (WT) littermate controls. However, we noticed a mild but significantly reduced body weight starting on week 11-12 of age for the males and on weeks 8 and 15 of age for the females *Axl KO* mice compared to wild-type control littermates (Supplementary Fig. 6A). Measurements of mRNA expression in iBAT and ingWAT in these mice, demonstrated significantly increased expression of genes that support thermogenesis in *Axl* KO as compared to WT controls, such as Ucp1, Dio2 and Cox8b as well as increased expression of the regulator of

mitochondrial biogenesis Pgc1-α and of the lipid-associated Elovl3 (Supplementary Fig. 6B). Next, we evaluated the response of *Axl KO* mice to a cold stimulus. Gene expression analysis in iBAT depots demonstrated significantly enhanced thermogenic capacity of *Axl* KO mice in response to two days of cold exposure, as evaluated by increased expression of several thermogenesis-associated genes including Ucp1, Pgc1-α and Cox8b as well as brown fat-enriched genes such as Dio2 and Elovl3 (Supplementary Fig. 6C). This effect was even

**Fig. 5 | Pharmacological inhibition of AXL receptor induces a pAKT-mediated upregulation of intracellular cAMP levels in response to β-adrenergic stimulation.** Mature iBAs were serum-depleted for 2-3 hours and acutely treated with AXL inhibitors, anti-AXL antibody, PDE inhibitors and/or GAS6 in the presence or absence of isoproterenol. **A** Transcription factor enrichment analysis in upregulated genes upon AXL receptor siRNA-mediated knockdown. **B** Intracellular cAMP levels at indicated time points after isoproterenol stimulation and simultaneous treatment with treatment with AXL inhibitor BMS-777607, or AXL agonist GAS6 (* Control vs. BMS; # Control vs. GAS6) ($n = 3$/group). **C** Intracellular cAMP levels in response to acute (15 min) treatment with BMS-777607 and simultaneous stimulation with increasing isoproterenol doses ($n = 3$/group). **D** Representative western blots of 3 independent experiments of phosphorylated PKA substrates. **E** Representative western blots of 3 independent experiments of phosphorylated CREB (pCREB) and ATF (pATF) after (60 min) isoproterenol stimulation and concomitant treatment with BMS-777607 and/or GAS6. HSP90 was used as a loading control. **F)** OCR (Seahorse assays) measurements in isoproterenol-stimulated iBAs upon concomitant treatment with BMS-777607 or DMSO control ($n = 5$/group). Arrow signifies the time of injection of isoproterenol and compound. **G** Intracellular cAMP levels of isoproterenol-stimulated mature iBAs in response to acute treatment with an AKT activator (left panel) and an AKT inhibitor (right panel), in the presence or absence of the BMS-777607 ($n = 3$/group). **H** Intracellular cAMP levels of isoproterenol-stimulated mature iBAs in response to acute treatment with a PI3K inhibitor compound (left panel) and a PTEN inhibitor compound (right panel), in

the presence or absence BMS-777607 ($n = 3$/group). **I** Representative western blots of two independent experiments of total AXL receptor expression in A549 cells treated for indicated times with anti-AXL antibody. siRNA-mediated knockdown of AXL receptor was used as positive control. γ-TUBULIN was used as loading control. **J** Representative western blots of three independent experiments of pAKT (S473) in response to acute treatment with anti-AXL antibody in mature iBAs. BMS-777607 was used as positive control and HSP90 was used as loading control. **K** OCR measurements in isoproterenol-stimulated (1uM) mature iBAs in response to treatment with anti-AXL antibody or IgG control ($n = 6$/group). Arrow signifies the time of injection of isoproterenol (Iso), IgG control or anti-AXL antibody (Ab). **L** Direct phosphodiesterase activity (relative values) in acutely isoproterenol-stimulated mature iBAs in response to treatment with anti-AXL antibody or IgG control ($n = 4$/group for PDE4, $n = 3$/group for all other conditions). **M** Gene expression analysis of Ucp1 and Pgc1-α in mature iBAs in response to pharmacological inhibition of PKA (PKAi), in the presence or absence of BMS-777607 ($n = 4$/group). N) Gene expression analysis of Ucp1 and Pgc1-α in mature iBAs in response to siRNA-mediated knockdown of Atf2 (siAtf2), in the presence or absence of BMS-777607 ($n = 4$/group). For all graphs, results are presented as average ± SEM. $*p < 0.05$, $**p < 0.01$, $***p < 0.001$, $****p < 0.0001$. For two-group comparisons (**C, L**), an unpaired two-tailed t-test was performed. For three or more group comparisons (**E, G, H, M, N**) one-way ANOVA was performed. Tukey test was applied to correct for multiple comparisons. For time-course comparisons (**B, F, K**) two-way ANOVA was performed. Statistically significant differences are annotated in graphs.

---

more pronounced than what we previously observed under regular room temperature housing conditions (Supplementary Fig. 6B vs. Fig. 6C). In addition, we could show that *Axl KO* mice presented with increased energy expenditure as compared to wild-type littermates, effect that was more pronounced under cold-exposure environmental conditions (Supplementary Fig. 6D).

Subsequently, we investigated the effect of HFD challenge on the metabolic phenotype of *Axl KO* mice. Strikingly, we noticed that *Axl KO* female and male mice demonstrated a dramatically reduced body weight and attenuated weight gain in response to HFD, as compared to the wild-type controls, an effect that was apparent already within the first 4 weeks of HFD and became even more pronounced after 15-25 weeks under the high-caloric food challenge (Fig. 6B and Supplementary Fig. 6E). These results suggest that *Axl KO* mice are resistant to diet-induced obesity. As we initially hypothesized that AXL receptor deletion would enhance brown fat activity, we performed microscopic H&E analysis in iBAT depots of *Axl* KO and WT littermates, which revealed reduced lipid accumulation and increased number of multilocular adipocytes in the iBAT depots of the *Axl KO* mice, suggesting improved brown fat functionality (Fig. 6C). These morphological changes were associated with reduced weight of iBAT depots in *Axl* KO mice (Fig. 6D). After 25-30 weeks on HFD, *Axl KO* mice presented with a reduced body weight, which was mainly attributed to reduced fat depot mass, while lean mass was unchanged between the two groups (Fig. 6D). This difference in body weight was not attributed to reduced food intake (Fig. 6D). We further characterized the physiology of this mouse model under HFD conditions by estimating energy expenditure of *Axl KO* and wild-type littermates, using metabolic cages to measure their oxygen consumption (VO2). We observed that *Axl KO* mice demonstrated significantly higher VO2 and energy expenditure, compared to controls (Fig. 6E). These changes were accompanied by increased expression of genes that support thermogenesis in iBAT depots of *Axl* KO mice, such as Ucp1, Pgc1-α, and Elovl3 (Supplementary Fig. 6F). Interestingly, *Axl* KO mice only showed a mild but not significantly improved insulin tolerance compared to the controls (Supplementary Fig. 6G). Accordingly, fasting blood glucose levels were only mildly but not significantly reduced in *Axl* KO mice (Supplementary Fig. 6G).

The discrepancy between improved weight gain and reduced fat mass and the absence of improvement in glucose tolerance, led us to hypothesize that AXL receptor knockout majorly affects lipid metabolism instead, effect concordant with the well-established role of

active brown fat to efficiently sequester circulating lipids[14]. To test this, we challenged *Axl* KO and WT controls by exposing them to a high-fructose/high-fat diet, in order to generate a diet-induced dyslipidaemia model with non-alcoholic fatty liver disease (NAFLD) and included chow diet-fed animals as negative controls. Indeed, this diet led to weight gain, increased circulating free fatty acids (FFAs), as well as a dramatic increase in total cholesterol and liver weight (Fig. 6F). As expected, *Axl* KO mice had significantly lower body weight in response to a high fat diet (Fig. 6F - left panel). Next, we measured circulating lipids and observed that *Axl* KO mice demonstrated lower levels of total cholesterol, and FFAs as compared to WT controls (Fig. 6F). In accordance with our previous HFD cohorts, fasting blood glucose was mildly but not significantly reduced. However, *Axl* KO mice demonstrated dramatically reduced liver weight compared to controls, suggesting protection against ectopic lipid accumulation (Fig. 6F). Taken together, these data suggest that deletion of AXL receptor switches brown fat activity toward an enhanced lipid-oxidative state, effect that is reflected by improved dyslipidaemia and reduced lipid accumulation in the liver.

As the *Axl* KO model is based on a whole-body ablation, we performed additional studies to convincingly demonstrate that the aforementioned effects are primarily mediated via enhanced brown fat function and are not attributed to other tissues. First, we placed HFD-fed *Axl* KO and WT mice in thermoneutral conditions (30 °C), in order to "switch off" brown fat function. Remarkably, the previously observed effect of AXL receptor deletion on reduced weight gain under standard room temperature housing conditions (Fig. 6G) was entirely abolished under thermoneutral conditions (Fig. 6H), indicative of the contribution of non-shivering thermogenesis and enhanced brown adipose tissue function in the reduced body weight of *Axl KO* animals. In addition, we employed a combination of pharmacological AXL inhibition, which demonstrated that both pharmacological AXL inhibition and *Axl* KO led to significantly reduced body weight in response to HFD, as compared to DMSO-treated or WT littermate controls, respectively. Notably, pharmacological AXL inhibition in *Axl* KO mice did not confer an additive effect in reduced weight gain (Fig. 6I) demonstrating the specificity of the pharmacological compounds. Similar to our previous observations, insulin tolerance of BMS-treated or *Axl* KO mice showed a trend towards improvement (Supplementary Fig. 6H).

We performed whole-tissue bulk RNA sequencing in iBAT depots of the HFD-exposed mice, analysis which demonstrated an extended

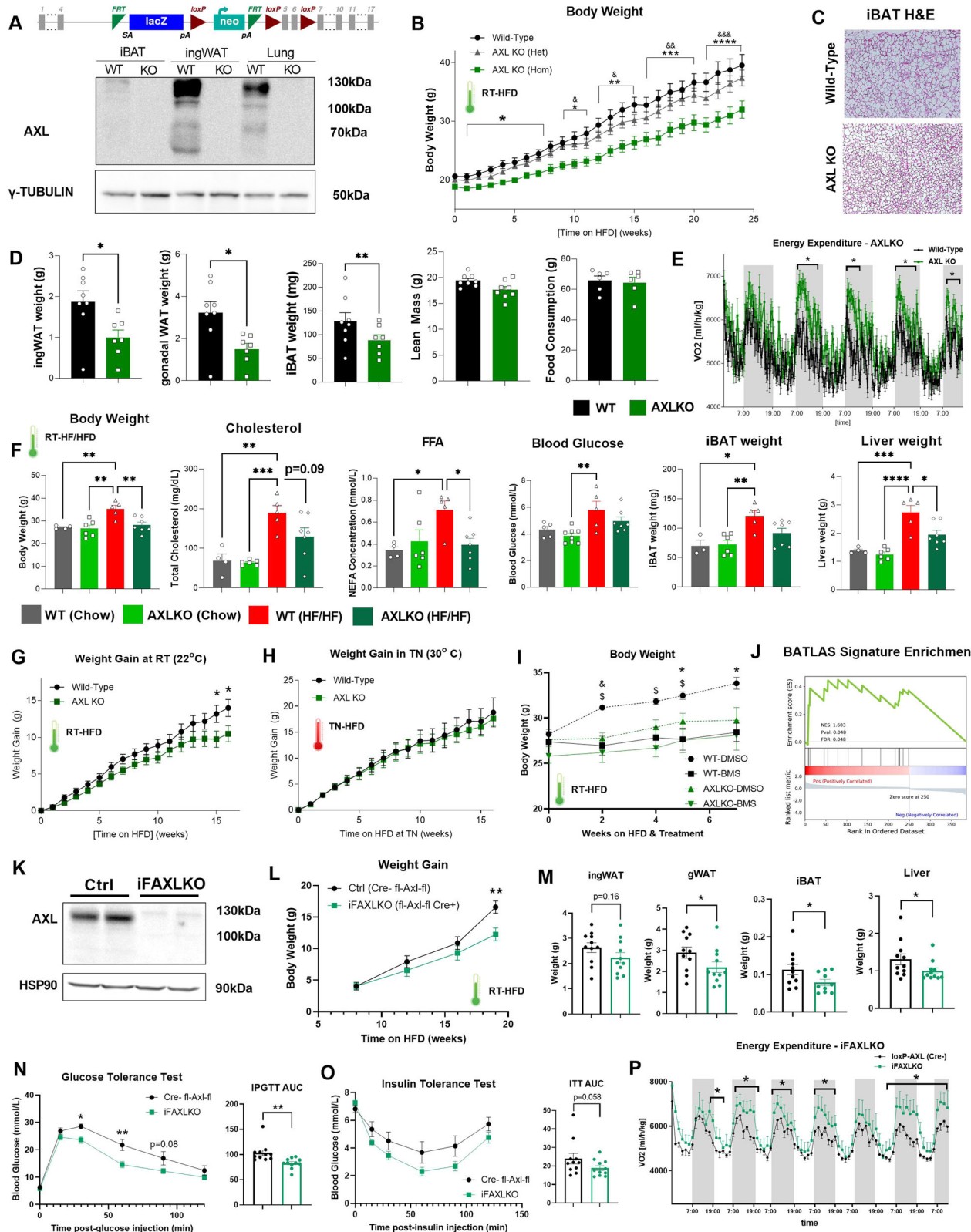

number of significantly differentially expressed genes. Subsequent bioinformatic analyses revealed that the differentially expressed genes of the iBAT depots of *Axl* KO mice demonstrated a significantly higher enrichment for the BATLAS[50] thermogenic gene expression signature (Fig. 6J). In addition, KEGG pathway analysis revealed that several pathways related to thermogenesis, PPAR signaling, oxidative phosphorylation, as well as pathways associated with regulation of lipolysis

in adipocytes and with fatty acid metabolism, elongation, and degradation appear to be significantly upregulated in the *Axl* KO mice, compared to wild-type littermate controls (Supplementary Fig. 6K). Overall, these data point towards an enhanced brown fat functionality in response to AXL receptor inhibition as well as brown fat activity-mediated effects in improvement of systemic metabolism in *Axl* KO mice.

**Fig. 6 | *Axl* KO and iFAXLKO mouse models are protected from diet-induced obesity (DIO). A** Transgenic cassette and subsequent global deletion of AXL receptor (upper) and representative western blot of three independent experiments of AXL receptor in tissues of global *Axl* KO (homozygous) mice (lower). **B** Weekly body weight curves of *Axl* KO homozygous (AXLKO Hom), *Axl* KO heterozygous (AXLKO Het) and WT female littermates on HFD challenge. HFD was initiated at 8 weeks of age, mice were acclimatized to room-temperature conditions ($n = 27, 29, 31$; respectively). $*p < 0.05$, $** p < 0.01$, $***p < 0.001$ and $****p < 0.0001$ (for WT vs. AXLKO Hom). $^\& p < 0.05$, $^{\&\&} p < 0.01$, $^{\&\&\&} p < 0.001$ and $^{\&\&\&\&} p < 0.0001$ (for AXLKO Het vs. AXLKO Hom), two-way ANOVA was performed. **C** Representative H&E staining images iBAT depots of *Axl* KO and WT littermates after 24 weeks of HFD. iBAT depots of all *Axl* KO and WT mice were imaged. **D** Measurements of inguinal WAT (ingWAT), gonadal WAT, and iBAT fat depot weights (after 24 weeks of HFD), lean mass (via EchoMRI – after 9 weeks of HFD), and food consumption measurements of *Axl* KO and WT littermate controls, after 24 weeks of HFD challenge (WT $n = 8$; *Axl* KO $n = 7$) (**E**) VO2 metabolic cage measurements in *Axl* KO and WT female mice 24 weeks after HFD challenge ($n = 6$/group). F Body weight, total cholesterol, free fatty acids (FFA), fasting blood glucose, iBAT weight, and liver weight of *Axl* KO and WT littermate controls, after 32 weeks of a high-fructose high-fat (HF/HF) diet challenge that was used to generate a diet-induced obesity and fatty liver accumulation mouse model (WT-chow $n = 4$; *Axl* KO-chow $n = 6$; WT-HF/HF $n = 5$; *Axl* KO-HF/HF $n = 6$). **G** Weekly weight gain of *Axl* KO and WT littermate controls, in response to 16-week HFD challenge at room temperature (RT) ($n = 25$ for *Axl* KO and $n = 30$ for WT). **H** Weekly weight gain of *Axl* KO and WT littermate controls, in response to 16-week HFD challenge, at thermoneutrality (TN) ($n = 10$/group). **I** Body weight of *Axl* KO and WT littermate controls after treatment with the pharmacological AXL receptor inhibitor BMS-777607 (BMS) or DMSO, in response to HFD challenge. **J** RNA-Seq analysis of iBAT depots from *Axl* KO and WT male mice after 25 weeks on HFD and subsequent enrichment analysis of the BATLAS signature score of the differentially expressed genes in *Axl* KO vs. WT mice ($n = 5$/group) **K**) Western blot of the mature fraction of ingWAT depots from control (*Axl*-floxed Cre-) or iFAXLKO (*Axl*-floxed CreERT2 +) mice of AXL receptor in adipocytes. **L** Weight gain curves of iFAXLKO and littermate control (lox-AXL-lox Cre-) female mice in response to HFD challenge ($n = 11$/group). **M** Weight of ingWAT, gWAT, iBAT, and liver in iFAXLKO and floxed-AXL-Cre- control littermates, after 20 weeks of HFD ($n = 11$/group). **N** Glucose tolerance test (glucose dose: 2 g/kg) and **O**) insulin tolerance test (insulin dose: 1IU/kg) in iFAXLKO and loxP-AXL-Cre- control littermates, after 20 weeks of HFD ($n = 11$/group). **P** VO2 metabolic cage measurements in iFAXLKO and littermate control (loxP-AXL (Cre-)) female mice 8 weeks after HFD ($n = 6$/group). For all graphs, results are presented as average ± SEM. $*p < 0.05$, $**p < 0.01$, $***p < 0.001$, $****p < 0.0001$. For two-group comparisons (**D, M**), unpaired two-tailed t-test was performed. For three or more group comparisons (**F**), one-way ANOVA was performed (Tukey test was applied to correct for multiple comparisons), and for time-course datasets (**E, G–I, L, N–P**) two-way ANOVA was performed. Statistically significant differences are annotated in graphs.

To define the adipocyte-specific contribution to these phenotypes in a more conclusive manner, we generated an inducible adipocyte-specific AXL knockout mouse model (fl-Axl-fl CreERT2+), hereby termed iFAXLKO (Fig. 6K) and characterized its metabolic phenotype in response to HFD. Similar to the constitutive whole-body *Axl* KO mouse model, iFAXLKO mice showed protection from diet-induced obesity by significantly reduced body weight (Fig. S6I) and weight gain (Fig. 6L) compared to their control (fl-Axl-fl CreERT2-) littermates (Fig. 6K). This was accompanied by reduced gonadal WAT (gWAT) and liver weights, indicative of lower fat mass and reduced ectopic lipid accumulation, respectively (Fig. 6M), as well as by reduced iBAT depot weight, indicative of reduced lipid content and enhanced oxidative activity of the brown fat (Fig. 6M). Furthermore, this reduced body weight corresponded to improved glucose tolerance and insulin sensitivity, as demonstrated by glucose tolerance test (Fig. 6N), and insulin tolerance test (Fig. 6O). In addition, iFAXLKO mice demonstrated increased energy expenditure (Fig. 6P) and showed a trend towards reduced cholesterol levels, despite an absence of dyslipidaemia (Supplementary Fig. 6J).

We performed whole-tissue bulk RNA sequencing in iBAT depots of iFAXLKO vs. Cre- control mice; differential gene expression analysis demonstrated an extended number of significantly upregulated and downregulated genes (Supplementary Fig. 6L). We specifically focused on cluster 2 - where most upregulated genes in iFAXLKO vs. Cre- control mice clustered together - and performed overrepresentation pathway analysis. Even though most enriched pathways were associated with extracellular matrix (ECM) organization, selected enriched pathways also included "Response to cAMP" and "GPCR Signaling" pathways, suggesting "sensitization" of the cAMP/PKA pathway, in accordance with our previously described mechanistic studies. Additional pathway analysis via Enrichr revealed "Fatty Acid Metabolism" and "Glycolysis" enriched pathways (Supplementary Fig. 6M). Taken together, these data suggest that adipocyte-specific *Axl* deletion confers resistance to diet-induced obesity, improved glucose metabolism, and leads to enhanced functionality of brown fat of the iFAXLKO mice. Overall, we show that both *Axl* KO and iFAXLKO mouse models are resistant to HFD-induced obesity and have an improved metabolic phenotype. These effects are associated with transcriptomic changes in the brown fat that support non-shivering thermogenesis, as well as improved adipose tissue functionality.

## Discussion

In the recent years, several pathways have been described that can stimulate and activate brown or beige fat, but their translation to humans is either underway or has not been successful, so far. For instance, FGF21 which has been shown to increase energy expenditure in rodents[51], also reduced body weight in humans[52]. However, whether these effects are due to the modulation of food intake or increased energy expenditure mediated by brown adipose tissue functionality remains elusive. Past attempts to use β3-adrenergic agonists (such as mirabegron) showed very moderate body weight and energy expenditure effects in humans[53], while a recent study argued that human BAT-mediated thermogenesis is driven by β2- and not by β3-adrenergic receptors[54]; targets which are likely limited in their applicability due to possible cardiovascular adverse effects[55]. Here, we describe a role of the AXL receptor in the functionality of brown and white adipose tissues, which can be harnessed to induce non-shivering thermogenesis, thus promoting a negative energy balance. We could demonstrate that pharmacological AXL receptor inhibitors increased energy expenditure, a phenotype that was associated with induced brown fat-mediated non-shivering thermogenesis. These observations render AXL receptor a promising pharmacological target that exerts the potential to stimulate non-shivering brown adipocyte-mediated thermogenesis and shift energy balance toward a negative state.

The differential AXL receptor expression between white and brown adipose tissue as well as the surprising pronounced and acute effect of β-adrenergic stimulation on the protein levels of AXL receptor are indicative of its putative role in thermogenesis. Our comprehensive time-course analysis concerning the regulation of AXL receptor both at the mRNA and protein level, indicates a potential post-transcriptional regulation of the receptor in response to isoproterenol stimulation. Whether this can be attributed to cleavage of the extracellular domain of AXL receptor or to internalization of the receptor and subsequent proteasomal degradation or to other degradation-mediating post-translational modifications, remains to be clarified[56]. Our data suggest that β-adrenergic stimulation may enhance AXL receptor cleavage in brown adipocytes, as isoproterenol induces levels of circulating AXL. Importantly, our data also suggest that endogenous brown adipose-derived GAS6 may be driving basal AXL receptor signaling. In fact, brown adipocytes secrete GAS6 in the culture medium and significant amounts of GAS6 are detected in the plasma in mice, while β-adrenergic stimulation leads to increased GAS6 levels in vitro and in vivo. This is also supported by our findings of basal levels of

phosphorylation of AXL receptor in the absence of GAS6 and insulin/FBS, as well as a reduction of pAXL in response to treatment with AXL inhibitors in the absence of any additional GAS6 stimulation. Similarly, downstream phosphorylation analyses showed reduced pAKT in response to AXL inhibitors in the absence of GAS6 stimulation, while phosphorylation of other RTKs was not reduced by the AXL inhibitor. Taken together, these findings point towards paracrine and endocrine mechanisms, through which AXL receptor is constitutively activated.

The role of AXL receptor on brown- and beige-fat-mediated thermogenesis and the associated underlying molecular mechanisms have not been explored. Taken together, our mechanistic data demonstrate that AXL receptor signaling fine-tunes the Insulin-AKT signaling pathway and that a moderately attenuated Insulin/IR-AKT signaling in fat depots promotes the transcriptional machinery that leads to enhanced browning and stimulated uncoupling thermogenesis. An important distinction must be made, though, between the acute (signaling to PDEs) and the chronic (requirement for adipogenesis and adipocyte differentiation) role of insulin and insulin signaling. Despite the fact that a complete loss of insulin signaling will impair the function of brown fat[17], a host of evidence suggests that mild attenuation of insulin/IR/AKT signaling in brown adipose tissue can lead to enhanced BAT-mediated thermogenesis whereas, on the other hand, excessive insulin can impair brown fat functionality. A recent study demonstrated that daily insulin injections in mice caused significantly reduced expression of UCP1 and PGC1-α in their iBAT[18]. Accordingly, mice with an inducible adipocyte-specific deletion of PTEN which resulted in increased phosphorylation of AKT and enhanced insulin signaling in their adipose depots, demonstrated significantly reduced expression of brown adipocyte-specific genes such as Ucp1, Pgc1-α and Prdm16 in their iBAT depots[57]. Inversely, deletion of one of the insulin alleles led to increased EE that was associated with enhanced browning of WAT, identifying circulating insulin as a critical negative regulator of Ucp1 expression[19]. As a member of the family of receptor tyrosine kinase proteins, the regulation of the phosphorylation of AKT upon agonism and/or antagonism of AXL receptor has been thoroughly characterized in several cell lines and tissues[58]. A number of elegant studies using adipocyte-specific knockout mouse models have provided insights on the role of AKT in brown fat function[21,22,59,60]. Taken together, these studies reveal the necessity of functional insulin/AKT signaling pathway for BAT. It should be noted, however, that in all aforementioned models, insulin signaling is completely abolished rather than moderately attenuated. Contrary a model with partial impaired insulin signaling, such as the Ai-IGF1RKO model exhibited increased expression of most thermogenic and brown fat-enriched genes in iBAT[17]. Along these lines, modest transgenic PTEN overexpression (PTEN[tg]) led to mild downregulation of the PI3K/AKT pathway, which was associated with enhanced iBAT thermogenic capacity[61], whereas adipose tissue specific AKT1 KO mouse model was protected against DIO and demonstrated enhanced BAT thermogenesis[22]. Similarly, treatment of brown and white adipocytes with PI3K inhibitors, which mitigated but did not completely ablate the PI3K/AKT pathway led to an increased expression of thermogenic and mitochondrial genes[61]. These results were confirmed by in vivo administration of PI3K inhibitors in obese mice and non-human primates, which led to reduced adiposity due to enhanced BAT-mediated thermogenesis[62]. Similarly, we demonstrate here that inhibition of AXL receptor signaling only moderately attenuates but does not completely block the activity of the insulin/AKT pathway and results in increased brown adipocyte functionality. Taken together, these results underscore the importance of separating the information on complete ablation vs. attenuation, which can elicit different effects.

Our work shows that the observed enhanced thermogenic program upon inhibition of AXL receptor signaling is partially mediated via increased FOXO1 activity. Previously, FOXO1 has been identified as a major transcription factor responsible for Ucp1 upregulation upon PTEN overexpression. Indeed, studies demonstrated enriched direct binding of FOXO1 on the Ucp1 and Pgc1-α promoters of PI3K inhibitor-treated brown adipocytes as well as reduced FOXO1 phosphorylation in the iBAT of the PTEN[tg] mouse model and upon PI3K inhibitor treatment, while a constitutively active FOXO1 resulted in enhanced BAT function[61,62]. Contradictory results have been reported on the role of FOXO1, with studies reporting that its deletion in adipocytes can lead to Ucp1 upregulation[63], however, these findings involved Akt-independent mechanisms and did not demonstrate Ucp1 down-regulation upon FOXO1 gain-of-function, suggesting that alternate mechanisms might be involved. The effect of our FoxO1 knockdown data suggests that FOXO1 is required for thermogenesis in brown adipocytes and that it mediates the upregulation of thermogenesis-associated genes induced by AXL inhibition. Importantly, BAT-specific mTORC2 loss was shown to deacetylate and activate FOXO1, with nuclear FOXO1 retention potentiating UCP1 expression[64]. Similarly, we demonstrate that inhibition of AXL receptor signaling reduces phosphorylation of FOXO1 which, in turn, promotes FOXO1 nuclear localization in brown adipocytes. Along these lines, we propose that this effect is exerted in an AKT-mediated manner, as shown by the moderate downregulation of the insulin/AKT pathway due to its crosstalk with the GAS6/AXL pathway, the subsequent attenuated FOXO1 phosphorylation, and the recapitulation of enhanced FOXO1 nuclear localization upon treatment with AKT inhibitors. Taken together, we propose that AXL receptor inhibition enhances thermogenic gene expression in a FOXO1-mediated manner in non-stimulated brown adipocytes.

Intriguingly, our studies suggest that under β-adrenergic-stimulated conditions (such as isoproterenol or cold exposure), enhanced thermogenesis in response to AXL receptor inhibition or deletion is primarily mediated via an AKT-PDE3/4 inactivation pathway and a subsequent stimulation of cAMP and ATF2. We propose that inhibition of AXL receptor signaling removes the brake of the thermogenesis-inducing β-adrenergic/cAMP pathway, by blunting the effect of the cAMP-degrading PDE3 and PDE4 enzymes in an AKT-mediated manner. Our rescue experiments using pharmacological AKT activators and PTEN inhibitors to ablate the AXL inhibitor-induced elevation of intracellular cAMP levels underscore the point that the observed increased cAMP levels are indeed attributed to reduced AKT phosphorylation. Remarkably, AKT and PI3K inhibitors phenocopying AXL receptor inhibitors with a small or absent additive effects on cAMP levels when used in combination, further support a mediation via the AKT signaling pathway. It is worth pointing out that these effects could also occur independently in discrete pools of AKT. However, as insulin is one of the most potent anabolic hormones and a major driver of AKT phosphorylation in FBS-depleted adipocytes, we suggest the existence of a simplified model where AKT is the central hub in which activities of IR and AXL receptor intersect. Further research will be required to investigate in detail the exact effects of AXL receptor inhibition on discrete AKT pools. Along the same lines, our results do not rule out that it is solely the activity of AKT which drives these effects. Lastly, it is important to point out that our data do not suggest that the effects are specific to the AXL receptor signaling pathway. In fact, other studies have shown that other RTKs can exert similar effects, such as the deletion of IFG1R[17].

More direct data of an AXL-PDE axis stem from our observations that stimulating AXL receptor using an anti-AXL receptor agonistic antibody specifically suppresses the activity of PDE3 and PDE4 and reduces cAMP levels, whereas PDE3 and PDE4 inhibitors mimic the effect of AXL inhibitors to increase cAMP levels. In accordance with our studies, adipose tissue-specific deletion of the PI3K subunit p110α leads to reduced fat accumulation, due to potentiation of β-adrenergic signaling via PDE inhibition[65]. Additionally, PDE4 inhibitor roflumilast leads to reduced weight gain, increased energy expenditure and improved glucose metabolism in DIO mice[66], while it improves glucose

tolerance in patients with T2D[67]. Similarly, a combination of PDE3 and PDE4 inhibition has been shown to induce Ucp1 expression and lipolysis in brown adipocytes, while PDE1/2/8 A inhibition did not exert such effects[68]. Recently it was shown that differently compartmentalized cAMP pools in brown adipocytes are differentially affected by β1/2- and β3-adrenergic receptors, with PDE3 and PDE4 inhibition contributing to both and majorly to β1/2-induced cAMP pool[69]. In light of the recent observations about the role of β2-adrenergic receptor in human BAT, these results suggest that AXL receptor inhibitors could exert even more pronounced effects in human BAT activation[54]. The observed phenotype of our mouse model system supports the mechanistic model of a crosstalk between the β-adrenergic and AXL receptor pathways, as increased thermogenic gene expression in iBAT, enhanced energy expenditure and weight loss in *Axl* KO mice are accentuated in lower (stimulated β-adrenergic signaling) compared to higher (attenuated β-adrenergic signaling) temperature conditions across all experiments.

Our *Axl* KO mouse model reveals a previously unappreciated biologically significant role of AXL receptor on body weight regulation which is in part mediated through its effects on enhanced brown adipose tissue functionality. Enhanced thermogenic transcriptional signature, reduced brown adipocyte size concomitant with reduced lipid content and increased Ucp1 expression in iBAT of *Axl* KO mice, provide convincing proof that the effect of AXL receptor deletion is BAT-mediated. Despite the absence of dramatically improved glucose tolerance - possibly due to a transient mild downregulation of AKT signaling pathway in other peripheral tissues - the major beneficial phenotypes of the *Axl* KO mouse model are primarily associated with weight loss and improved lipid metabolism, as demonstrated by improved circulating lipids and reduced ectopic lipid accumulation. Indeed, recent human studies have demonstrated that increased BAT mass is associated with cardiometabolic health[70]. Taking into consideration the prevalent dyslipidemias in the presence of obesity and the absence of pharmacotherapy against non-alcoholic or metabolic fatty liver disease (NAFLD or MAFLD), this phenotype renders AXL receptor deletion an interesting therapeutic model. These data are in line with the *Gas6* KO mouse model, which exhibits significantly reduced fat mass accumulation upon HFD-induced obesity[36]. Intriguingly, *GAS6* and *AXL* polymorphisms have been associated with adiposity and insulin resistance in adolescents, providing human relevance to the role of AXL receptor signaling in metabolic disease[31]. Our whole-body *Axl* KO data cannot entirely rule out that resistance to diet-induced obesity and increased energy expenditure may be attributed to other tissues, as AXL receptor is broadly expressed. Along the same lines, an AXL receptor gain-of-function approach may confer adverse effects in vivo. For instance, macrophages of the cleavage-resistant AXL mouse model (Axl^CR/CR) have enhanced phospho-STAT1 and HIF-1α signaling as well as elevated glycolysis and IL-1β secretion, suggesting a role of AXL receptor toward proinflammatory metabolic reprogramming[71].

Our adipocyte-specific AXL receptor deletion experiments strongly support even further an adipose-mediated contribution toward improved metabolic phenotype of *Axl* KO in the fat. Indeed, deletion of AXL receptor specifically in adipocytes (iFAXLKO) has a similarly improved body weight phenotype, confirming that the improved metabolic phenotype of the *Axl* KO mouse model is mostly attributable to adipose tissue. Remarkably, iFAXLKO mice demonstrate significantly improved glucose tolerance, highlighting the fact that the absence of significantly improved glucose tolerance in *Axl* KO mice may be attributed to other peripheral tissues and that adipocyte-specific AXL receptor deletion does not impair - rather it enhances – peripheral tissue glucose uptake.

In summary our data identify AXL receptor as a regulator of brown adipocyte activation and functionality. In the context of brown adipocyte signaling, AKT acts as a continuum of gradually enhanced or attenuated activity and not as an "on-off" switch. Pharmacological treatments, genetic manipulations, or environmental stimuli that moderately attenuate the insulin/AKT pathway while maintaining its activation above a presumable "dysfunctional threshold", induce the appropriate transcriptional machinery to promote enhanced brown adipocyte functionality, while genetic or pharmacological manipulations that inhibit the AKT pathway below this "dysfunctional threshold" lead to impairment of the metabolic processes of the cells and subsequent attenuated functionality as well as reduced expression of thermogenic genes. On the other hand, over-stimulation of the insulin/AKT pathway that can surpass an upper "overly active threshold" of AKT activity can repress the function of key brown-adipocyte transcriptional regulators and lead to a reduced expression of thermogenic genes. We propose that inhibition of AXL receptor signaling promotes a "fine tuning" of this pathway (Fig. 7) which can be leveraged to modulate brown adipocyte activity and thereby increase energy expenditure to prevent weight gain in obesity.

## Methods

### Cell culture of A549 cell line
Immortalized A549 cell lines were isolated from the lung tissue of white, 58-year-old male with lung cancer and purchased from ATCC (Cat.No. CCL-185). A549 cells were cultured in RPMI-1640 medium (ThermoFisher Scientific, Gibco, Cat.No. 61870036) supplemented with 10% Fetal Bovine Serum (FBS) and 1% Penicillin/Streptomycin.

### Cell culture of adipogenic cell lines
Immortalized mouse brown and white pre-adipocyte cell lines derived from the stromal-vascular fraction of either interscapular brown or epididymal white adipose tissue of newborn or young mice were obtained and cultured as previously described[39,45]. In brief, cells were cultured in high glucose (25 mM) DMEM containing 10% Fetal Bovine Serum (FBS) and 1% Penicillin/Streptomycin. All lines were maintained below confluency by splitting every 2-3 days. Cells were differentiated in rat collagen-coated plates using differentiation induction medium (DIM).

### Immortalized adipocyte differentiation and stimulation
To induce adipogenic differentiation, the cells lines were grown to confluence on collagen-coated tissue culture plates and maintained for 2 additional days. Immortalized brown preadipocytes were induced by stimulation with 115 μg/ml IBMX, 1 μM dexamethasone, 20 nM insulin, 125 μM indomethacin and 1 nM T3 in full medium (defined as Day 0 of differentiation). After 48 h the medium was exchanged for full medium plus 20 nM insulin and 1 nM T3. Cells were maintained in this medium until sampling for analysis. For white adipogenesis, the cells were induced for 48 h in full medium containing the same adipogenic cocktail without indomethacin and T3. Full medium containing 200 nM insulin was provided for the remaining time of the differentiation process. To stimulate UCP1 expression for final read-out, the cells were incubated in full medium with 1 μM isoproterenol for 6-8 h before fixation or lysis.

### Cell culture – hMADS cells
Human multipotent adipose-derived stem (hMADS) cells were obtained from Dr. Ez-Zoubir Amri's laboratory (University of Nice, France) and cultured as previously described[46]. In detail, hMADS cells were grown in low glucose DMEM supplemented with 15 mM HEPES, 10% FBS, 2mM L-glutamine, 1% Penicillin/Streptomycin and 2.5 ng/ml recombinant human FGF-2 (Peprotech). The medium was changed every other day and FGF-2 was omitted after cells reached confluence. Differentiation of 48 hours post-confluent cells was induced (day 0) by adipogenic medium (DMEM/Ham's F12 media containing 10 μg/ml Transferrin, 10 nM insulin and 0.2 nM triiodothyronine) supplemented with 1 μM dexamethasone and 500 μM isobutyl methylxanthine (IBMX)

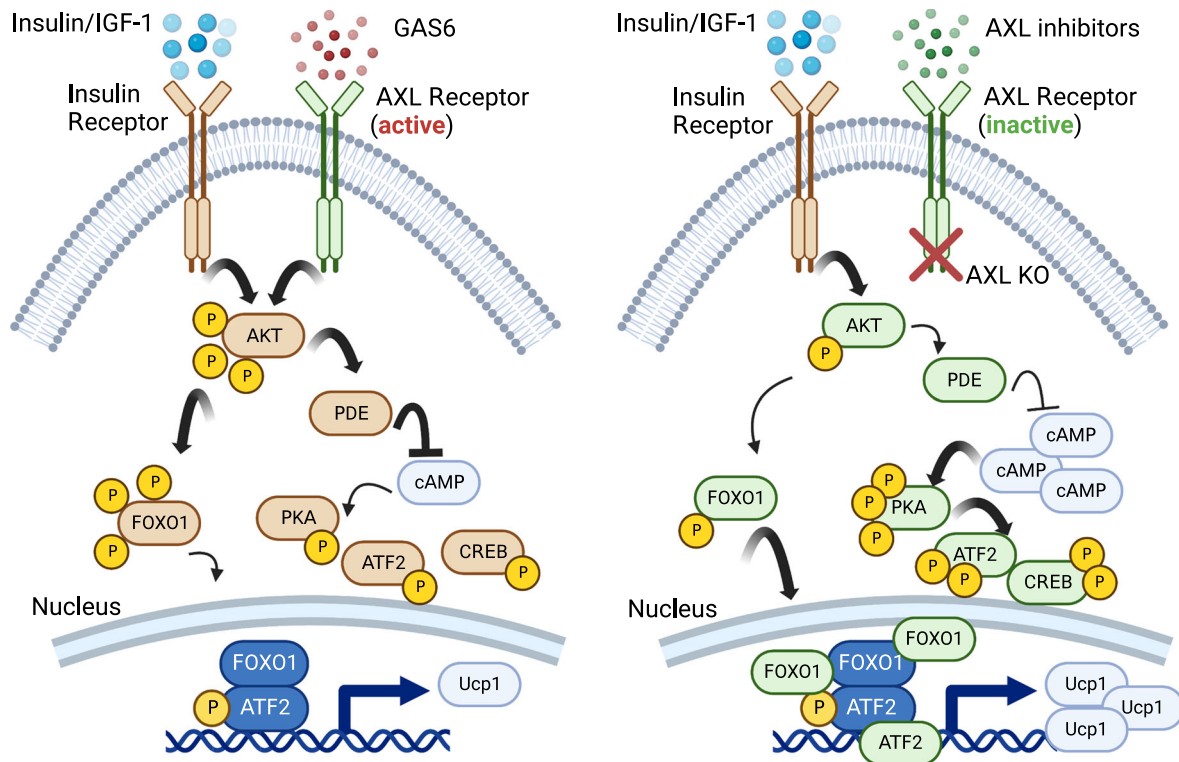

**Fig. 7 | The AXL/AKT/PDE/cAMP pathway regulates thermogenesis in brown adipocytes.** Current mechanistic view of the crosstalk between Insulin-IGF-1/Insulin Receptor and GAS6/AXL Receptor pathways and subsequent induction of a thermogenic phenotype in an AKT-cAMP-PKA manner (created with BioRender).

and from day 2 to 9, cells were cultured in adipogenic medium containing 100 nM rosiglitazone. Cells were kept in culture until day 18 in the absence of rosiglitazone to obtain mature white adipocytes. To obtain brown adipocytes, cells were exposed to an additional rosiglitazone (100 nM) pulse between days 14 and 18. To investigate the effect of pharmacological AXL inhibitors on the activation of brown adipocytes, treatment was performed between day 14 and 18 in combination with rosiglitazone. To knockdown AXL receptor genes, 50 nM siRNA against AXL or non-targeted (N.T.) controls (Microsynth) were delivered into mature adipocytes on day 13 using Lipofectamine RNAiMAX (Invitrogen). After 24 hours, the transfection medium was replaced by fresh adipogenic medium containing 100 nM rosiglitazone. Brown adipocytes were cultured until day 18, when cellular respiration was determined or cells were harvested for RNA and protein analysis.

### siRNA-mediated knockdown

For siRNA-mediated knockdown, Dharmacon smartPOOL on-target PLUS RNA pools (consisted of 4 different siRNAs) were used for reverse transfection. Subsequently, the 4 siRNAs were separately tested for their knockdown efficiency. After picking the most efficient siRNA, 100-250 nM of siRNA (i.e. 20-50 nM final working concentration of siRNAs) were incubated for 20 min in 1.5% Lipofectamine RNAiMAX diluted in OptiMEM in a ratio of 15 μL RNAiMAX/1 mL of OptiMEM. Subsequently, the siRNA-RNAiMAX-OptiMEM solutions were mixed with 400.000 (12-well plates) or 200.000 (24-well plates) or 50.000 (96-well plates) trypsinized mature cells in 800 μL (12-well plates), 400 μL (12-well plates) or 80 μL (96-well plates) full growth medium and plated on collagen-coated plates. siRNA-containing medium was replaced by full growth medium 24 h hours post lipofection. 48-72 hours post-siRNA transfection, cells were harvested for RNA/protein extraction or were fixed for immunofluorescent analysis or were used for OCR measurements. In the stimulated assays, cells were stimulated with isoproterenol for 6-8 hours before harvesting.

### Immunofluorescent staining of cultured cells

For determination of brown and white adipocyte differentiation, cells were first washed in PBS twice for 5 min and fixated in 4% paraformaldehyde in PBS for 15 min. After an additional PBS washing step, the cells were incubated in the following staining cocktail for 20 min: 2 μM Hoechst, 100 ng/μl LD540, 5 μM Syto60 in PBS. After two PBS washing steps, the cells were covered light protected and stored at 4 °C for microscopical analysis[72]. After analyzing the first staining, an additional UCP1 immunohistochemical staining was performed. The samples were further permeabilized and fixed by incubating them in pre-cooled 5% acetic acid in ethanol for 10 min at −20 °C. Cells were washed in PBS at room temperature twice and blocked/permeabilised for 90 min in blocking buffer (0.05% Triton X-100, 5% BSA, in PBS). Primary UCP1 antibody (Pierce) was incubated 1:500 overnight, washed, and an Alexa488- coupled secondary was incubated for 1 h. After counterstaining with Hoechst, the samples were washed three times in PBS and covered light-protected and stored for image-based analysis. Image acquisition for immunofluorescence was performed by Operetta automated microscope (Perkin Elmer). Immunofluorescence image analysis was performed by Harmony v3.5 (Perkin Elmer).

### RNA extraction, cDNA synthesis, quantitative RT-PCR

Total RNA was extracted from tissues or cells using Trizol reagent (Invitrogen) according to the manufacturer's instructions. DNase treatment was performed using DNase I (NEB) to remove traces of genomic DNA. Reverse transcription was performed to generate cDNA library by using the High Capacity cDNA Reverse transcription kit (Applied Biosystems), with 1 μg of RNA. Quantitative PCR was performed using the Fast SYBR Green qPCR Mastermix (Applied Biosystems) on a ViiA7 real-time PCR system (Applied Biosystems). Real-time PCR was analyzed by ViiA7 Ruo v1.2.3 (ThermoFisher). Relative mRNA concentrations normalized to the expression of 36B4 or TBP (mouse tissues and cell culture) were calculated by the ΔΔCt method. Primer sequences are found in key resource table.

## Measurement of intracellular cAMP concentrations

Pre-iBAs were differentiated to mature iBAs and cAMP assays were performed during days 6-8 of their differentiation. In brief, cells were washed with DMEM (25 mM glucose) without FBS and serum-depleted with DMEM (25 mM glucose) without FBS, 2 hours prior to iso-proterenol stimulation and/or compound treatment. Whenever pre-treatment was required, this was performed during the 2 h serum-depletion period. Stimulation with isoproterenol or forskolin and treatment with compounds was acutely performed for 5 min – 30 min. After the stimulation period, cells were rapidly washed with PBS and subsequently processed according to manufacturer's instructions. The Enzo cAMP ELISA kit was utilized to process cells after stimulation and to measure intracellular cAMP concentrations (Enzo, ADI-900-067). An aliquot of the cell lysate was used for BCA protein assay, for protein quantification and normalization of the cAMP concentrations. Bio-chemical colorimetric assays were detected by SynergyMix plate reader (BioTek). Colorimetric assays were analyzed by Gen5 v3.08 (BioTek).

## Animal work

All authors of the study who participated in mouse experiments have complied with ethical regulations for animal testing and research. All animal procedures in this study were approved by the cantonal ethics committee of the veterinary office of the Canton of Zürich. The sample size was determined based on previous experiments in our lab and similar studies reported in the literature. All mice used for the experiments were either female or male as indicated, housed 3–4 lit-termates per cage in individually ventilated cages at standard housing conditions (22 °C, 12 h reversed light/dark cycle, dark phase starting at 7am, 40% humidity), with ad libitum access to chow (18% proteins, 4.5% fibers, 4.5% fat, 6.3% ashes, Provimi Kliba SA) or high fat diet (HFD) (60% (kal%) fat, diet no. 3436, Provimi Kliba SA), as indicated, and ad libitum access to water. Health status of all mouse lines were regularly monitored according to FELASA guidelines. At the end of the study, animals were euthanized singly in carbon dioxide atmosphere.

## Generation of the whole-body Axl KO and adipocyte-specific Axl KO mouse models

*Axl* KO mice were obtained by blastocyst injections of two clones (H04 and D04) of modified ES cells (Axl^tm1a(EUCOMM)HMGU) acquired from EUCOMM by performing pronuclear injections of ES cells containing the transgenic "knockout-first" cassette[73] which was inserted between exons 4 and 5 of the *Axl* gene, using classic homologous recombination techniques. The cassette contains mouse splicing acceptor (SA), LacZ, and neomycin-resistant gene (Neo). This initial allele (tm1a) generates a null allele via splicing to the LacZ trapping element. Thus, the Axl^tm1a(EUCOMM)HMGU allele is a non-expressive form, due to alternative splicing of Axl exon1-4 to SA in the trapping cassette, thus disrupting the Axl transcript. In detail, modified ES cells were thawed, expanded and subsequently injected into blastocysts which were implanted into foster C57Bl6 wild-type mothers/carriers until the generation of chi-meras, which carried a germ-line mutation. Level of chimerism in the offspring of the foster mothers was determined through qPCR as well as visually, based on the agouti coat color, gene which was also encoded in the modified ES cells. The selected chimeras (level of chi-merism was more than 90%) were further bred with C57Bl6 wild-type mice for the generation of heterozygous animals. The final global *Axl* KO mouse model was generated by breeding heterozygous x hetero-zygous animals, breeding which resulted in the mendelian generation of 25% *Axl* KO – 25% wild-type – 50% heterozygous mice.

Adipoq-CreERT2 animals were created by bacterial artificial chromosome (BAC) cloning of CreERT2 into the RPCI-24-69M4 BAC vector (BACPAC, Oakland, CA, USA) and pronuclear injection, as we have previously described[74]. For the generation of the iFAXLKO mice, heterozygous transgenic *Axl* KO animals were crossed with Flipase (FLP) homozygous animals for the removal of the transgenic cassette between the FRT sites, resulting in mice heterozygous for loxP-Axl-loxP sites. Offspring of this breeding were crossed with the Adipo-QERT2Cre mouse line, until the generation of loxP-Axl-loxP homo-zygous mice (FLP-) AdipoQERT2Cre+ (heterozygous). For the generation of the final adipocyte-specific AXL KO mouse model, the following breeding was performed: AdipoQERT2Cre+ (heterozygous) loxP-Axl-loxP (homozygous) x AdipoQERT2Cre- loxP-Axl-loxP (homo-zygous). The offspring of the latter breeding resulted in the expected mendelian ratio of 50% AdipoQERT2Cre+ loxP-Axl-loxP and 50% Adi-poQERT2Cre- loxP-Axl-loxP mice. For the induction of the adipocyte-specific AXL knockout, 6-8 week-old mice were gavaged with 2 mg/kg hydroxy-tamoxifen for two consecutive days and placed on a high-fat diet. Tamoxifen gavages were repeated every 3 weeks for two con-secutive days, to avoid the possibility of AXL receptor presence due to newly differentiated adipocytes.

## Pharmacological treatment mouse experiments

For in vivo pharmacological studies, C/57Bl6J wild-type mice were obtained by the Charles-Rivers laboratories at 4-6 weeks of age and were subsequently housed for 2 weeks in our animal facilities prior to the initiation of the experiments, in order to be acclimatized to the new housing conditions. For the in vivo pharmacological experiments, 8-10 weeks old male C57Bl6 mice (Charles River) were placed on HFD for 6-8 weeks and subsequently subjected to BMS-777607 (5-10 mg/kg/day) or DMSO control treatment (dissolved in sunflower oil for gavaging or in HFD powder for food-mediated administration) for 14 consecutive days by oral gavage or through diet-mediated administration. BMS-777607 for the in vivo studies was purchased from Sigma-Aldrich (Cat.No.: AMBH303C671F-250MG). At the end of the 2-week treatment, mice were fasted for 4 hours and iBAT, ingWAT and liver were dissected, snap-frozen in liquid nitrogen and maintained frozen in −80 °C until further analysis. Plasma exposure of the com-pound was confirmed by LC-MS analysis.

AXLKO mice were housed in groups of 3-4 mice (for males) or 3-5 mice (for females) per cage in individually ventilated cages (IVC) at standard housing conditions (23 °C, 12 h reversed light/dark cycle, dark phase starting at 7am), with *ad libitum* access to chow diet and water. For the HFD challenge, 8-10 weeks old AXLKO (homozygous), AXLKO heterozygous and wild-type littermate controls were placed under *ad libitum* HFD in groups of 3-4 mice per cage in IVC at standard housing conditions and body weight was recorded once per week, on the same day and time of the week. For the thermoneutrality challenge, 8-10 weeks old AXLKO and wild-type littermate mice were placed into chambers which maintained the housing conditions at 30 °C, 40 humidity and 12 h reversed light/dark cycle, dark phase starting at 7am. Body weight was recorded once per week, on the same day and time of the week. For the cold-exposure challenge, AXLKO and wild-type lit-termate mice were housed into chambers which maintained the housing conditions at 8 °C, 60% humidity and 12 h reversed light/dark cycle, dark phase starting at 7am.

## Body composition measurements

Live mice body composition was measured with a magnetic resonance imaging technique (EchoMRI 130, Body Composition Analyzer, Echo Medical Systems). Fat and lean mass was analyzed using Echo MRI 14 software. Mice were fasted for 4 hours before measurement.

## Indirect calorimetry – metabolic cages

Indirect calorimetry measurements were performed with the Pheno-master (TSE Systems) according to the manufacturer's guidelines and protocols. $O_2$ and $CO_2$ levels were measured for 60 s every 13 min continuously with Phenomaster (TSE Systems). Energy expenditure analysis was performed by Phenomaster software v5.6.5 (TSE-systems). The respiratory quotient was estimated by calculating the ratio of $CO_2$ production to $O_2$ consumption. Animals were single-caged and

acclimated to the metabolic cage for 48 hours prior metabolic recording. Locomotor activity, food and water intake were monitored throughout the whole measurement.

## Cellular respiration

For measurement of cellular respiration, immortalized murine brown adipocytes were cultured on collagen-coated 10 cm cell culture dishes. Differentiating adipocytes were trypsinized and replated on day 5 at a density of 30.000 cells per well and allowed to recover and mature for 48–72 hours before treatment. Since hMADS cells grow in a monolayer, they were differentiated directly on collagen-coated 96-well Seahorse cell culture microplates. On the day of the experimental treatment, adipogenic medium was replaced with unbuffered XF Assay Medium (pH 7.4, Seahorse Bioscience) supplemented with glucose (1 g/L, Sigma), 2 mM sodium pyruvate (Invitrogen) and 2 mM GlutaMax (Invitrogen). The oxygen consumption rate (OCR) was measured using the Extracellular flux analyzer XF96 (Agilent Seahorse). For the measurement of the different components of respiration (mitochondrial, non-mitochondrial and uncoupling respiration), test compounds were sequentially injected to obtain the following working concentrations: 1 μg/ml Oligomycin, 0.5 – 1 μM isoproterenol, (0.5 mM cAMP for hMADS), 1 μg/ml FCCP, 3 μM Rotenone with 2 μg/ml Antimycin A. OCR levels (pmol/min) were normalized to protein amount per well (μg protein). Non-mitochondrial respiration was subtracted to obtain basal, uncoupled, stimulated uncoupled and maximal mitochondrial respiration. For the acute stimulation with the compounds (AXL/PDE inhibitors), cells were pre-treated for 2-3 hours before the Seahorse assay with the respective compounds (dissolved in XF Assay Medium) and, subsequently, isoproterenol (0.5 μM) was co-injected with the compounds for the measurement of baseline and isoproterenol-stimulated OCR. Cell respiratory analysis was done by Wave 2.6.0 (Agilent Seahorse). Data were exported in GraphPad Prism for visualization and analysis.

## Protein extraction and Western blot

Tissue samples and in vitro differentiated adipocytes were homogenized in RIPA buffer (50 mM Tris pH 7.4, 150 mM NaCl, 2 mM EDTA, 1.0% Triton X100, 0.5% sodium deoxycholate) supplemented with protease (Complete, Roche) and phosphatase (Halt phosphatase inhibitor cocktail, Thermo Fisher) inhibitor cocktails, as well as with 1 mM PMSF (for the detection of protein phosphorylation). Lysates were cleared by centrifugation at 12,000 g for 15 minutes at 4 °C. Protein concentration of the supernatants was determined by BCA Assay (Thermo Scientific). Western Blotting was carried out following standard procedures. Equal amount of proteins (30-40 μg) were separated on 8-12% SDS-polyacrylamide gel, transferred to a nitrocellulose membrane (Bio-Rad) and blotted for the following antibodies: UCP1 (ThermoFisher, Cat# PA1-24894, RRID: AB_2241459, 1:1000), γ-TUBULIN (Sigma-Aldrich, Cat# T-5326, RRID: AB_532292, 1:10,000), HSP90 (Cell Signaling, Cat# 4877, RRID: AB_2233307, 1:1000), phospho-CREB Ser133 (Cell Signaling, Cat# 9198, RRID: AB_2561044, 1:1000), phospho-PKA substrates (Cell Signaling, Cat# 9624, RRID:AB_331817, 1:1000), phospho-AKT Thr308 (Cell Signaling, Cat# 13038, RRID: AB_2629447, 1:1000), phospho-AKT Ser473 (Cell Signaling, Cat# 4060, RRID: AB_2315049, 1:1000), phospho-Foxo1 (Thr24)/FoxO3a (Thr32) (Cell Signaling, Cat# 9464, RRID:AB_329842, 1:1000), FOXO1 (Cell Signaling, Cat# 2880, RRID:AB_2106495, 1:1000), anti-mouse HRP secondary (Merck, Cat# 401253, RRID: AB_437779, 1:5000), anti-rabbit HRP secondary (Merck, Cat#401393, RRID: AB_10683386, 1:5000), anti-mouse Alexa Fluor 488 secondary (Thermo Fisher, Cat# A-11029, RRID:AB_2534088, 1:2000), anti-rabbit Alexa Fluor 488 secondary (Thermo Fisher, Cat# A-21206, RRID:AB_2535792, 1:2000), AXL (Abcam, Cat# ab215205, RRID: AB_2924328, 1:1000), AXL (ThermoFisher, Cat# PA5-106118, RRID: AB_2853927, 1:1000), PGC1-a (Millipore Sigma, Cat# AB3242, RRID: AB_2268462, 1:1000), phospho-Insulin Receptor (Y972) (Abcam, Cat#ab5678, RRID: AB_305045, 1:1000), phospho-Insulin

Receptor (Y1361) (Abcam, Cat#ab60946, RRID: AB_943587, 1:1000), phospho-IRS1 (ThermoFisher, Cat#PA5-114593, RRID: AB_2899229, 1:1000), phospho-AXL (Tyr702) (ThermoFisher, Cat# PA5-64862, RRID: AB_2662770, 1:1000), phospho-STAT3 (Ser727) (Cell Signaling, Cat#9134, RRID: AB_331589, 1:1000), phospho-STAT5 (Tyr694) (Cell Signaling, Cat#4322, RRID: AB_10544692, 1:1000), phospho-PDGFRA (Y742) (Abcam, Cat#ab5452, RRID: AB_304899, 1:1000), phospho-PDGFRB (Y857) (Abcam, Cat#ab62367, RRID: AB_2162635, 1:1000). Chemiluminescent signal of the HRP-conjugated secondary antibodies (anti-mouse or anti-rabbit purchased from Merck or Calbiochem or Cell Signaling, dilution 1:5000) was visualized by the Image Quant system (GE Healthcare Life Sciences). Quantification of the intensity of the bands was performed by ImageJ v 1.53e (NIH).

For the pAXL (Tyr702) measurement, the high-throughput Sally Sue technique was used, according to the manufacturer's protocol. In detail, equal amount of undiluted protein lysates (concentration >0.2 mg/mL) were loaded and primary pAXL (Tyr702) antibody (1:15 dilution) was used for the detection of the phosphorylated AXL receptor. The assay was performed following the instructions of the kit (Sally Sue version 1, ProteinSimple, Bio-Techne). pAXL (Tyr702) was quantified using the Compass software. pAXL was defined as 138 kDa with 10% of tolerance and Gaussian Fit was used for area calculation. A signal/noise ratio of 10 or less was defined as a "not significant peak", in which cases the area was set to zero.

## Phosphodiestarase (PDE) activity assay

PDE activity was measured in mature iBAs (Days 6-8 of differentiation) using a colorimetric PDE Activity Assay Kit (Abcam – Cat.No.: ab139460), as previously described[65]. Briefly, mature iBAs were co-incubated with isoproterenol (2 uM) and the agonistic AXL receptor antibody or IgG control for 15 minutes. Subsequently, brown adipocytes were lysed in a buffer containing 20 mM Hepes-NaOH (pH 7.4), 0.5 mM EDTA, 2 mM MgCl2, 0.1% Triton X-100, 0.5 mM DTT, 1 mM EGTA, 1 μM microcystin-LR and protease inhibitors. Lysates were briefly sonicated, desalted using Zeba Spin desalting columns, 7 kD molecular weight cut-off (Thermo Scientific, cat. no. 89882), and equilibrated in PDE assay buffer. Desalted lysates (15 μg protein) were assayed for phosphodiesterase activity using the colorimetric kit from Abcam (cat. no. ab139460) according to the manufacturer's instructions. This PDE Abcam kit measures total PDE activity and the isoform-specific PDE activity is specifically attributed to each one of the seven distinct isoforms via the utilization of specific PDE inhibitors that we employed. Using specific PDE inhibitors, the reduction of activity to the total activity accounts for the specific activity of the individual PDE isoform.

## Pharmacological inhibition of AXL receptor tyrosine kinase and phosphodiesterases (PDEs) and other compounds

Pharmacological compounds BMS-777607 (Cat.No.: S1561), Bemcentinib (R428) (Cat.No.: S2841) and NPS-1034 (Cat.No.: S7669) were purchased from Selleckchem and dissolved in DMSO to a stock concentration of 10 mM. For AXL receptor inhibition in cultured cell lines, serial dilutions were performed (in DMSO and/or in DMEM) to achieve the required final concentrations in the complete maintenance DMEM (+Insulin/T3). DMSO concentration in the final complete DMEM was maintained at 0.1% v/v. PDE1, PDE2, PDE4, PDE8, and PDE10 inhibitors were kindly provided by Boehringer Ingelheim. Cilostazol (PDE3 inhibitor) was purchased from Millipore-Sigma (Cat.No.: C0737) and dissolved in DMSO to a stock concentration of 10 mM. PKA inhibitor H89 was purchased from Tocris (Cat.No. 2910). PI3K, PTEN, AKT, and FOXO1 inhibitors were purchased from Selleckchem.

## Intracellular cyclic AMP (cAMP) measurements

Cells were FBS-depleted for 2 hours prior to the cAMP assays in high-glucose DMEM medium. Treatment (isoproterenol and/or compounds)

was performed in high-glucose DMEM medium. Prior to assay, cells were briefly washed 1x with PBS. Subsequently, intracellular cAMP concentration was measured using a cAMP ELISA kit (ENZO, Cat.No.: ADI-901-067A), following manufacturer's instructions. Colorimetric assay was detected by SynergyMx plate reader (BioTek).

## RNA Sequencing and Analysis
RNA from iBAT and ingWAT was quality-checked by a TapeStation instrument (GE). All samples had a RNA integrity number (RIN) > 8. rRNA was depleted, and purified RNA was used for the preparation of libraries using the TruSeq RNA sample preparation kit (Illumina). Samples were sequenced on a HiSeq 4000 HT instrument. RNA-seq sequences were trimmed using Trim Galor and mapped to the mouse GRCm38 genome assemblies using hisat2 (v2.1.0). Transcripts were defined using the Ensemble annotations over protein-coding mRNAs. Differential expression analysis of mapped RNA-seq data was performed using DESeq2[57] and EdgeR. Significantly different transcripts were called with significance <0.05 after Benjamimi and Hochberg correction and minimum mean differential expression of twofold. Gene ontology (GO) analysis was performed using the g:profiler software. Expressed genes had a log2(RPM) value > 0. PCA was performed using the R prcomp package with default parameters.

AXL expression in human, mouse and rat normal tissues has been profiled using previously published RNASeq data sets (see Array Express submissions E-MTAB-6081 and E-MTAB-1733, respectively). Accordingly, raw FASTQ files have been processed with the RNASeq data analysis pipeline as described in Schlager et al.[75] to obtain FPKM (fragments per kilobase of exon model per million mapped reads) normalized gene expression levels.

## Tissue harvest
Animals were euthanized singly in carbon dioxide atmosphere. All tissues were carefully dissected, weighed, snap frozen in liquid nitrogen and maintained frozen in −80 °C until further processing. Popliteal lymph nodes were carefully removed from inguinal adipose depots for all gene and protein expression analyses.

## Histology and image analysis
Adipose tissues were excised, fixed in fresh 4% paraformaldehyde (Sigma) in PBS (Gibco; pH 7.4) for 48 h at 4 °C and then embedded with paraffin. Standard hematoxylin and eosin staining was performed on rehydrated 4-micron paraffin sections. Slides were dehydrated and covered with coverslip by resin-based mounting. Tissue sections were visualized by Axiophot microscope equipped with AxioCam MR (Zeiss).

## Axl receptor expression in tissue panels
AXL receptor expression in human normal tissues has been profiled using previously published RNASeq data sets (see Array Express submissions E-MTAB-6081 and E-MTAB-1733, respectively). Accordingly, raw FASTQ files have been processed with the RNASeq data analysis pipeline as described in[75] to obtain FPKM (fragments per kilobase of exon model per million mapped reads) normalized gene expression levels.

## Statistical analyses and quantifications
For in vivo studies, littermates randomly assigned to treatment groups were used for all experiments. Sample sizes were determined on the basis of previous experiments using similar methodologies. The animal numbers used for all experiments are indicated in the corresponding figure legends. All animals were included in statistical analyses, and the investigators were blinded. All cell culture experiments were performed with 2-3 technical replicates for RNA and protein analysis, 5-6 replicates for measurement of cellular respiration and independently reproduced at least 3 times. Results are reported as

mean ± SEM. Two-tailed unpaired Student's t-test was applied on comparison of two groups. One-way ANOVA analysis and Tukey test for correction of multiple comparisons were applied on comparisons of multiple (three or more) groups. Two-way ANOVA was applied for comparisons of two or multiple groups across several time-points. All statistical analyses were performed using Graphpad Prism 9 software. Statistical differences are indicated as * for $p < 0.05$, ** for $p < 0.01$, *** for $p < 0.001$ and **** for $p < 0.0001$.

## Reporting summary
Further information on research design is available in the Nature Portfolio Reporting Summary linked to this article.

## Data availability
RNA-Sequencing datasets (hMADS transcriptomic datasets after ablation of AXL receptor, iBAT transcriptomic datasets of Axl KO vs. WT mice and iBAT transcriptomic datasets of iFAXLKO vs. WT mice) used in this study have been deposited and are available in GEO under accession number GSE231471. All data that support the findings of this study are included in this paper as its Supplementary Information files. Source data are provided with this paper.

## Code availability
Differential expression analysis of the bulk RNA-Seq data was performed following the standard tutorial of the R packages DeSeq2 and edgeR.

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

## Acknowledgements

We would like to thank Elke Kiehlmann for paraffin blocks and H&E adipose histology preparation. We would also like to thank SCOPEM and Functional Genomic Center Zurich (FGCZ) for next generation RNA Sequencing analysis. We thank Heike Neubauer and Bradford Hamilton from Boehringer Ingelheim for their valuable input and technical support. RNA-Seq analysis was performed by Functional Genomics Center Zurich. The schematic overview of AXL receptor inhibition mechanism in Fig. 7 was created using BioRender.

## Author contributions

V.E., B.H., H.N., and C.W. designed the study; V.E., L.D., B.H., and C.W. supervised the experiments; V.E., L.D., M.B., W.S., L.B., L.G.S., H.D., A.P., S.K., C.H., and C.M. performed the experiments; V.E., and C.H. performed image analysis; V.E., W.S., and H.D. performed RNA-Sequencing experiments, E.S., A.G., and U.G. performed all bioinformatics analyses; B.H., and H.N. provided resources; V.E. generated whole-body Axl KO knockout; V.E., and L.D. generated adipocyte-specific (iFAXLKO) knockout; V.E., and C.W. wrote the manuscript; all authors reviewed and edited the manuscript.

## Competing interests

The authors declare no competing interests.

## Inclusion and ethics statement

The paper adheres to the Nature Portfolio journals' authorship policy.
