## [Peer Review File · Nature Communications]

Inhibition of AXL Receptor Tyrosine Kinase Enhances Brown Adipose Tissue Functionality in miceREVIEWER COMMENTS

Reviewer #1 (Remarks to the Author):

NCOMMS-21-31486

The ms “Inhibition of AXL Receptor Tyrosine Kinase Enhances Brown Adipose Tissue Functionality” studies the role of AXL in human and murine brown adipocytes. For the in vivo studies global as well as tissue-specific knockout models were used. The authors show that ablation of AXL in adipocytes results in increased whole body energy expenditure and a significant reduction in diet-induced obesity. Moreover, the authors examine in detail the signaling pathway linked to AXL signaling in adipocytes. Overall this is a very interesting paper with novel findings. However, the following points should be addressed before publication:

General comments:

- Several labs have shown that insulin and IGF1 are essential for the development and function of BAT and WAT. The Kahn lab (Sakaguchi) has clearly shown that loss of ins/igf signaling leads to decrease of ucp1 in BAT and lipodystrophy with a reactive increase in ucp1 in WAT. This should be clearly stated in the introduction.
- One should differentiate between the acute (insulin PDE link) versus chronic (differentiation) effects of insulin in adipocytes, which should be emphasized and might explain the differences in the literature.
- The impressive in vivo phenotype should be further highlighted in the ms; perhaps some in vitro data could be moved to the supplement.
- AXL and other receptor tyrosine kinases activate jak/stat signaling, which has been shown to inhibit brown adipocyte differentiation. Could this – at least in part - be an explanation for the positive effects observed after AXL inhibition in the adipocytes (Figure 2)?
- Figure 4: FOXO localisation after insulin and gas6 treatment should be shown
- Figure 6: The iFAXLKO model is very interesting. The authors should please show more details for iFAXLKO (e.g. glucose tolerance, histological analysis).
- Figure 7: This scheme is a bit crowded; please focus on the most important connections and highlight them.

Minor points:

Line 686-689: “Interestingly, iBAT and ingWAT of Axl KO male mice demonstrated significantly enhanced thermogenic capacity, as evaluated by increased expression of several thermogenic genes including Ucp1, Pgc1- α and Cox8b as well as brown fat—enriched genes such as Dio2 and Elovl3(Fig. S6E). This effect was even more pronounced than what was observed under regular housing conditions (Fig. S6C).” The first sentence does not explain the housing conditions; thus, the second sentence is hard to understand.

Reviewer #2 (Remarks to the Author):

The main finding reported in the manuscript by Efthymiou et al is that pharmacological or genetic inactivation of the receptor tyrosine kinase Axl increases the activity of brown adipose tissue and results in significant reduced weight gain upon high fat diet feeding.

While the Gas6/TAM family receptor signalling system has been implicated in regulation of metabolism before, substantial mechanistic insights have been lacking. The present manuscript provides novel insights in the metabolic effects of this signalling system and highlights their translational potential in diseases associated with energy imbalance (obesity and type 2 diabetes).

The manuscript provides evidence for differential expression of Axl between white and brown adipose tissues. It also shows that inhibition of Axl enhances cAMP signalling through modulation of Akt activity and its downstream target PDE. The demonstration that the activity of Axl can affect cAMP levels is novel. The manuscript presents experiments in cell based models as well as phenotyping of energy metabolism in mice with constitutive global deletion of Axl and also in mice with inducible adipose tissue specific deletion of Axl.

The experiments are well designed and the interpretation of the data appropriate with a few exceptions that require clarification:

-Foxo1 is highlighted as a key mediator of the observed phenotypes, but this interpretation is essentially based on data reported by a previous study (PMID: 22405073) demonstrating that Foxo1 can bind on the PGC-1a or UCP-1 promoters. However, other than CHIP data presented in that paper, no additional evidence that Foxo1 is important for UCP-1 transcriptional control have been presented. Neither the present manuscript provides any mechanistic evidence that the observed phenotypic effects are dependent of Foxo activity. In this regard the attribution of phenotypic effects to Foxo transcriptional activity is rather speculative. Essentially, Foxo nuclear/cytoplasmic partition in the way tested in the manuscript is one more read-out of PI3K/Akt pathway activation. Given the effect of Axl inhibition on cAMP/PKA signalling demonstrated in the manuscript, it seems more likely that induction of UCP-1 expression is mediated through CREB/ATF activated in the adrenergic pathway. In fact, in the context of reduced Akt activity other mechanisms of thermogenic gene induction have been reported before such as potentiation of adrenergic signalling through reduction of PDE activity (PMID: 30948720).

-In a way, it is somewhat paradoxical that the effects of Axl inhibition are attributable to BAT activation, the tissue with the lowest expression of Axl. Since Axl is more highly expressed in iWAT, have the authors looked at inguinal WAT depots for signs of browning upon administration of Axl inhibitors?

-Fig. 5I The authors report that an antibody against Axl turned out to activate Axl signalling on the basis of increased Akt phosphorylation. But has the phosphorylation of Y702 been tested at least in early time points of antibody addition? Increases in pAkt could be through antibody cross-reactivity with other receptors.

Fig 5L (and Suppl. Fig. 5K) How was PDE activity specifically attributed to each one of the seven distinct isoforms indicated in the histogram?

Fig. S5J The use of the pan-PDE inhibitor IBMX to make the argument that Axl inhibition does not have an additive effect is probably not very useful. Treatment with IBMX would massively increase cAMP levels even in the absence of adrenergic stimulation. With the increase caused by IBMX being one or two orders of magnitude higher than that caused by treatment with Axl inhibitor, any additive effect by BMS would probably be masked.

Experimental procedures are described in good detail, but some more clarity is required in the description of Axl mouse gene targeting protocols:

- Description of AXL KO is somewhat confusing. "AXLKO mice were obtained by pronuclear injections of two clones (H04 and D04) of modified ES cells...". It is not clear where pronuclear injections were performed (or was it meant to be blastocyst injections?). Also, the schematic of the gene targeting vector (which should not be called 'transgenic cassette' as it creates confusion with generation of transgenic rather than gene targeted) shown in Fig. 6A is somewhat confusing with missing exons and lines connecting not continuous exons. Some more clarity needs to be provided and importantly to specify whether exons 4 and 5 were deleted (through recombination of the loxP sites in the ES cells or in the mouse through breeding with a germline Cre strain).
- A reference should be provided for the AdipoQERT2Cre mouse line used in the study. Also, the efficiency of the inducible AXL inactivation in the adipose tissue is a rather important information which is missing and a western blot for AXL protein levels in the adipose tissues of conditionally deleted mice should have been shown.
- The term 'congenital' is normally associated with human disease. I don't think I have seen it used in the context of mouse gene targeting. 'Constitutive' knockout is the term normally used to describe this type of gene targeting.

Minor Comments

- Experimental Procedures. The 'Sally Sue' method is quoted for measuring pAXL. 'Sally Sue' presumably refers to an instrument manufactured by Protein Simple (not specified).
- Fig. 1C histogram third bar labelled 'scWAT' presumably should be 'scBAT'.
- Fig. 1H Dose and length of treatment with CL316,243 are not specified in the figure legend.
- Fig. 2A Key of NPS doses colour coding incorrect.
- Fig. 2A, B Although specified in the legend it would be helpful if the figure also indicated that cells are stimulated with isoproterenol in A and non-stimulated in B.
- Fig. 2D While the two blots have different loading controls (HSP90 and tubulin) both histogram axes are labelled as 'UCP1 intensity /HSP90'
- Fig. 4E – D What is the difference between panels E and D?
- p. 27 "AXL receptor inhibition increases intracellular cAMP levels through moderate mitigation of the Insulin/AKT/PDE pathway". Was this meant to be "inhibition" instead of "mitigation"?
- Fig. 5B Although mentioned in the legend, it would be clearer if Y axis label was marked as 'isoproterenol-stimulated cAMP (pmol/mL/protein)'
- Fig. 5E Western blot data would be better presented as a histogram of average signal intensities.
- Suppl. Fig.1K legend "Direct phosphodiesterase activity (absolute values are shown)". Was it meant to be relative values?
- Suppl. Fig. 5D the time points (5 and 15 min) are not indicated in the graph.
- Suppl. Fig. 5I What is the time point of cAMP measurement?
- Suppl. Fig. 5J Isoproterenol, BMS,IMBX doses and time point are not specified.
- Suppl. Fig. 5M Are cAMP levels produced by different doses of PDE4i statistically significant?
- Suppl. Fig. 6H and 6I What was the sex of mice tested?
- Suppl. Fig 6K This graph is probably better to present together with that of Fig. 6H.

- Suppl. Table 1 Does a dash (-) indicates complete lack of cross-reactivity or no testing?
- p35, line 733 “and lean mass (lower)”. Should read “right” instead of “lower”
- p39, line 841: “in lower compared to higher conditions”. Presumably ‘temperature’ is missing.
- Several graphs and schematics have too small fonts or low resolution which make them illegible.

Reviewer #3 (Remarks to the Author):

Efthymiou et al., demonstrates a functional role of AXL inhibition in enhancing BAT thermogenesis and protection from HFD-induced obesity. They use pharmacological inhibition, siRNA and finally mice with genetic ablation of Axl to demonstrate enhanced UCP-1 and PGC-1a transcription and perform thermogenesis and HFD-weight gain studies. b-AR coupled Gs activates AC, which produces cAMP. cAMP activates PKA for thermogenesis. PDEs can negatively regulate cAMP and reduce thermogenesis. The authors postulate that GAS6-AXL dependent PI3K-PDK1-AKT axis not only excludes FOXO1 allowing PPAR γ access, but also activates PDEs and inhibits thermogenesis. Overall, the concept is interesting and surprising. AXL is more well known for its functional role in phagocytosis, anti-inflammatory signaling and promoting tumor EMT and migration/invasion.

Data is presented showing that AXL is broadly expressed including in adipose tissue. It is more prevalent in WAT relative to BAT thus giving rise to the notion that it might have a role in ‘beige’-ing of WAT. Immortalized brown adipocytes were stimulated with isoproterenol. Within 4 hrs, Axl mRNA was downregulated. Protein levels reduced by 2 hr. In vivo experiments also demonstrate reduced AXL. Similar results were observed in vivo with cold exposure or b-AR agonists.

AXL inhibitors increase UCP1, PGC1a and C/EBP in brown adipocytes and enhance brown-like phenotype in mature white adipocytes. It is indicated that the other TAM receptors or MET or RON are not expressed in these cells. AXL siRNA knockdown also enhances UCP1 and PGC1a. FOXO1 is translocated. A set of signaling data is presented indicating that GAS6 enhances AXL \sim P and pAKT \sim P. It is suggested that AXL inhibition reduces insulin-induced AKT \sim P.

Finally, the authors employ Axl KO to demonstrate increased UCP1 and PGC1a, enhanced thermogenic capacity and resistance to HFD-induced weight gain. Inducible, adipocyte-specific Axl deletion was also performed. It showed results similar to Axl KO.

While the data is interesting, some improvements are required in terms of quality, rigor and controls.

Major comments

1. The inconsistent pattern in AXL immunoblots throughout the manuscript is of grave concern (see specific comments).

2. AXL activation typically occurs when GAS6 bridges the receptor with phosphatidylserine (PtdSer). GAS6-PtdSer interaction is mediated by GLA domain of GAS. Is PtdSer-GAS6-AXL axis induced or constitutive?

Commercially, GAS6 is available in full-length or GLA-less forms. Ideally, both forms of GAS6 should be

included in the experiments to demonstrate GAS6-dependent effect on AXL. Minimally, full-length GAS6 should be used.

3. Conceptually, it is not apparent why Insulin-PI3K-PDK1-AKT axis is subthreshold and not sufficient for PDE activation (in the context of thermogenesis) and why is it necessary to invoke the idea of a threshold level of AKT activation that requires AXL signaling pathways? Furthermore, many RTK and other signaling pathways activate PI3K-PDK1-AKT axis. Can any of these other signals function cooperatively to exceed the threshold?

4. One plausible explanation may be that cold induces GAS6 or PtdSer exposure along with norepinephrine release from sympathetic neurons. This should be tested.

Specific comments

Figure 1. The band for AXL in immunoblots is not consistent in the panels. Were different antibodies used? Especially panel H is problematic. It would be better to include the molecular weight marker in the immunoblot.

Activation of AXL leads to its cleavage at an extracellular residue and shedding of sAXL.

Is AXL being cleaved following activation? What is the AXL epitope in Fig 1? Is there a corresponding increase in soluble AXL?

Figures 2 and 3. Strength includes use of pharmacological inhibitors and siRNA experiments.

In figure 2 panels A and B, letters a-d are used above the graphs to likely indicate p values while * is used in the legend. Can non-normalized values be shown to better represent the isoproterenol-induced upregulation?

Importantly, what is driving the basal AXL signaling? GAS6 should be included in these experiments to demonstrate ligand-dependent effect of AXL on transcriptional regulation.

What may be the physiological stimulus for GAS6 production and/or PtdSer exposure?

In Figure 3, similarly, GAS6 stimulation should be included for effect on UCP1 and other genes, as well as on OCR.

Panel A in this figure may be confusing to the readers since normalized expression is shown. TYRO3 and MERTK absolute expression appears much reduced in comparison to AXL (higher Ct in supplementary figure).

Since Axl KO and Axl f/f mice are available to the group, the pharmacological inhibition experiments can be complemented by genetic KO experiments and by showing a lack of further effect on genetically deleted lines (to rule out off-target effects). In this context, GAS6 stimulation experiments on knockout lines will add greatly to the specificity of the effects.

In Figure 4, BMS+GAS6 condition should be included for AKT~P. Quality of Figure 4E and F is poor and it is very difficult to clearly see FOXO1 nuclear exclusion. The major point is that AXL is driving some

nuclear exclusion even in the absence of insulin. Perhaps this point can be better made by including GAS6 in the absence and presence of insulin and by including nuclear export inhibitor as positive control along with the conditions already included.

There are also conceptual questions about this part. The conclusion that AXL inhibition reduces insulin-stimulated AKT~P is not evident from the data. There are possibilities that are not eliminated by the experimental design. AXL is well-known to drive AKT~P when bound to its ligand GAS6. Thus, AXL inhibition would be expected to reduce AKT~P. In parallel, insulin/IGF signaling stimulates AKT~P. However, these effects can happen independently in discrete pools of AKT.

One may also wonder that if this is simply a threshold issue, what is the actual threshold? PI3K? PDK1 activity? AKT~P?

More importantly, an explanation is required for specificity of the effect. Insulin-AKT is known to regulate PDEs via phosphorylation. Why is this axis not sufficient in the context of thermogenesis and why is it necessary to invoke AXL? Many RTK and other signaling pathways activate PI3K-PDK1-AKT axis. Why is thermogenesis the exclusive context wherein these signaling pathways additively function to establish the threshold?

Figure 5 is weak. It would be extremely important to use AKT inhibitors to reverse GAS6 effects (molecular and functional).

The use of GAS6 here is useful. It would be best to be consistent in using GAS6 throughout the experiments reported in the manuscript.

The PTEN inhibitors are known to be non-selective (PMID: 25446882). Constitutively active PI3K should be used to validate these findings. Similarly, constitutively active and kinase inactive AKT constructs should be used.

Again, the AXL band is inconsistent in panel I.

Figure 6 is a highly significant and important figure. However, there are concerns. Why is a AXL band present in lung of KO with CST antibody? Was Axl mRNA levels tested in the KO and iKO?

There is a published Axl CR/CR mouse (PMID: 33529176) where AXL signaling should be enhanced. It is tempting to speculate what would happen with regards to this mouse and thermogenesis.

In Supplementary Figure 3D, the names of most differentially genes could be included.

Since the knockout mouse lines are novel, they should be thoroughly characterized for genomic deletion (including sequencing), mRNA and for protein expression. Transcriptional profiling would also be useful.

REVIEWER COMMENTS

Reviewer #1 (Remarks to the Author):

NCOMMS-21-31486 – Response Letter to the Reviewers

We would like to thank the reviewer for the comments and for raising these important points. We have performed various additional experiments as outlined below to address these points.

General comments:

-Several labs have shown that insulin and IGF1 are essential for the development and function of BAT and WAT. The Kahn lab (Sakaguchi) has clearly shown that loss of ins/igf signaling leads to decrease of ucp1 in BAT and lipodystrophy with a reactive increase in ucp1 in WAT. This should be clearly stated in the introduction.

We thank the reviewer for this comment. In our revised introduction and discussion, we highlight now the published data on complete ablation of the IGF/IR signaling.

-One should differentiate between the acute (insulin PDE link) versus chronic (differentiation) effects of insulin in adipocytes, which should be emphasized and might explain the differences in the literature.

We thank the reviewer for raising this point. We have made this more clear in the manuscript and we have added a section to our revised introduction and discussion, summarizing the studies that show differences between chronic and acute effects of insulin in adipocytes.

-The impressive in vivo phenotype should be further highlighted in the ms; perhaps some in vitro data could be moved to the supplement.

We thank the reviewer for raising this point, indeed the in vivo phenotype we observed in the AXL KO and iFAXLKO mouse models is impressive and needs to be further dissected. To that end, we have substantiated the data on the whole-body (global) AXLKO mouse model, adding measurements on glucose tolerance (ITT, fasting blood glucose) and lipid metabolism (cholesterol, FFAs) (New Figures: Figure 6F, 6I and Suppl. Figure 6G, 6H). Moreover, we have added to our revised results section a more detailed characterization of the inducible adipocyte-specific iFAXLKO mouse model, including glucose and insulin tolerance tests, cholesterol measurements, as well as bulk RNA Seq gene expression analysis. We have rearranged the figures to include these new findings and to improve clarity. Overall, we conclude that AXLKO mice are resistant to diet-induced obesity, demonstrate an increased energy expenditure and are protected from dyslipidemias and ectopic liver fat accumulation. In addition, we now show that iFAXLKO mice are resistant to diet-induced obesity, have increased energy expenditure, improved glucose tolerance, and reduced liver fat accumulation (New Figures: Figure 6K, 6L, 6M, 6N, 6O and Suppl. Figure 6J, 6L, 6M).

-AXL and other receptor tyrosine kinases activate jak/stat signaling, which has been shown to inhibit brown adipocyte differentiation. Could this – at least in part - be an explanation for the positive effects observed after AXL inhibition in the adipocytes (Figure 2)?

We thank the reviewer for this excellent suggestion. We tested the effect of AXL pharmacological inhibitors in brown adipocytes on pSTAT3 and pSTAT5 (key substrates downstream of JAK) but we observed no significant

alteration of the JAK/STAT signaling. We therefore conclude that the effect is not mediated via this pathway. We have added the data to the revised manuscript (New Figure: Suppl. Figure 4H) and added a point on this to the discussion.

-Figure 4: FOXO localisation after insulin and gas6 treatment should be shown

FoxO1 localization is now shown in Figure 4 for all conditions (Figure 4E and 4F and Suppl. Figure 4D and 4E).

-Figure 6: The iFAXLKO model is very interesting. The authors should please show more details for iFAXLKO (e.g. glucose tolerance, histological analysis).

To address this suggestion, we utilized two approaches: 1) We performed bulk RNA-Seq analysis in iBAT adipose depots from the iFAXLKO mice which are now shown in Suppl. Figure 6L and Suppl. Figure 6M. 2) We performed the following additional metabolic and expression analyses: a. Body weight in response to HFD and weight of fat depots (ingWAT, gWAT, iBAT) and liver. b. Insulin tolerance test (ITT) and glucose tolerance test (IPGTT) analyses, c. Circulating lipid (cholesterol) measurements, and d. Axl protein expression analysis (western blot) in the mature fraction from fat (ingWAT) adipose tissues. Results are shown in Figure 6 K-O. We conclude that inducible adipocyte-specific Axl receptor deletion protects against diet-induced obesity, and leads to increased energy expenditure, reduced weight of fat depots and liver, and improved glucose tolerance. These effects are accompanied with transcriptomic changes in iBAT that are enriched for cAMP signaling.

-Figure 7: This scheme is a bit crowded; please focus on the most important connections and highlight them.

We have changed the scheme in Figure 7 and we now specifically highlight the pathways that were investigated in the present studies.

-Minor points:

-Line 686-689: "Interestingly, iBAT and ingWAT of Axl KO male mice demonstrated significantly enhanced thermogenic capacity, as evaluated by increased expression of several thermogenic genes including Ucp1, Pgc1- α and Cox8b as well as brown fat—enriched genes such as Dio2 and Elovl3(Fig. S6E). This effect was even more pronounced than what was observed under regular housing conditions (Fig. S6C)." The first sentence does not explain the housing conditions; thus, the second sentence is hard to understand.

We have changed the phrasing of this sentence accordingly, in order to specify the housing conditions.

Reviewer #2 (Remarks to the Author) - Response

The main finding reported in the manuscript by Efthymiou et al is that pharmacological or genetic inactivation of the receptor tyrosine kinase Axl increases the activity of brown adipose tissue and results in significant reduced weight gain upon high fat diet feeding.

While the Gas6/TAM family receptor signalling system has been implicated in regulation of metabolism before, substantial mechanistic insights have been lacking. The present manuscript provides novel insights in the metabolic effects of this signalling system and highlights their translational potential in diseases associated with energy imbalance (obesity and type 2 diabetes).

The manuscript provides evidence for differential expression of Axl between white and brown adipose tissues. It also shows that inhibition of Axl enhances cAMP signalling through modulation of Akt activity and its downstream

target PDE. The demonstration that the activity of Axl can affect cAMP levels is novel. The manuscript presents experiments in cell based models as well as phenotyping of energy metabolism in mice with constitutive global deletion of Axl and also in mice with inducible adipose tissue specific deletion of Axl.

The experiments are well designed and the interpretation of the data appropriate with a few exceptions that require clarification:

-Foxo1 is highlighted as a key mediator of the observed phenotypes, but this interpretation is essentially based on data reported by a previous study (PMID: 22405073) demonstrating that Foxo1 can bind on the PGC-1 α or UCP-1 promoters. However, other than ChIP data presented in that paper, no additional evidence that Foxo1 is important for UCP-1 transcriptional control have been presented. Neither the present manuscript provides any mechanistic evidence that the observed phenotypic effects are dependent of Foxo activity. In this regard the attribution of phenotypic effects to Foxo transcriptional activity is rather speculative. Essentially, Foxo nuclear/cytoplasmic partition in the way tested in the manuscript is one more read-out of PI3K/Akt pathway activation. Given the effect of Axl inhibition on cAMP/PKA signalling demonstrated in the manuscript, it seems more likely that induction of UCP-1 expression is mediated through CREB/ATF activated in the adrenergic pathway. In fact, in the context of reduced Akt activity other mechanisms of thermogenic gene induction have been reported before such as potentiation of adrenergic signalling through reduction of PDE activity (PMID: 30948720)

We thank the reviewer for raising this important point. Initially, we need to clarify that our studies highlight FoxO1 as a key mediator of the observed phenotypes in the non-isoproterenol stimulated conditions. As pointed out by the reviewer, we too think that in the isoproterenol-stimulated conditions, the elevated intracellular cAMP levels is the main driver of our phenotype, possibly mediated via PKA, CREB, and ATF2. We made necessary additions to our discussion in order to highlight this important distinction between mediators of the non-stimulated vs. isoproterenol-stimulated phenotypes.

To further support our hypothesis that the non-isoproterenol stimulated phenotype is mediated via FoxO1, we performed two different additional experiments: First, we used pharmacological FoxO1 inhibitors in combination with AXL inhibitors, and measured gene expression. As shown on Figure 4F, pharmacological inhibition of AXL increases thermogenesis-associated genes (as previously shown), FoxO1 inhibitor blocks thermogenic gene expression, whereas the combination of both inhibitors also abrogates the AXL inhibitor effect.

In addition, we performed siRNA mediated knockdowns (KD) of FoxO1 in the presence and absence of AXL inhibitors. As shown in Figure 4G, siRNA-mediated KD of FoxO1 dramatically reduces the expression of Ucp1 and Pgc1 α in iBAs. This effect cannot be rescued by the AXL inhibitor BMS, suggesting that FoxO1 is required for the observed upregulation of Ucp1 and Pgc1 α in response to Axl inhibition.

We do agree with the reviewer that potentiation of the beta-adrenergic signaling through inhibition of PDEs is likely one of the main mechanisms that AXL inhibition induces thermogenesis, we have made sure to include this reference in our discussion part.

-In a way, it is somewhat paradoxical that the effects of Axl inhibition are attributable to BAT activation, the tissue with the lowest expression of Axl. Since Axl is more highly expressed in iWAT, have the authors looked at inguinal WAT depots for signs of browning upon administration of Axl inhibitors?

Indeed, we observed effects not only in iBAT but also in ingWAT. In our pharmacological studies, we observe that the strongest effect (largest fold-expression upregulation) of the AXL inhibitor BMS in vivo is actually on Ucp1

mRNA (Figure 2F) and protein (Suppl. Figure 2F) levels in inguinal WAT. We have clarified this in the manuscript and added a section to the discussion to highlight this point.

-Fig. 5I The authors report that an antibody against Axl turned out to activate Axl signalling on the basis of increased Akt phosphorylation. But has the phosphorylation of Y702 been tested at least in early time points of antibody addition? Increases in pAkt could be through antibody cross-reactivity with other receptors.

We thank the reviewers for this very important point. In fact, we performed acute IF experiments in vitro using both AXL inhibitors and an activating AXL antibody, and we do show that AXL antibody activates both AXL receptor phosphorylation and AKT phosphorylation, whereas AXL inhibitors exert the opposite effects. The new IF data has been now inserted in Figure 4B.

-Fig 5L (and Suppl. Fig. 5K) How was PDE activity specifically attributed to each one of the seven distinct isoforms indicated in the histogram?

We apologize for this lack of information. PDE activity has been specifically attributed to each one of the seven distinct isoforms, using the respective kit from Abcam. This has now been detailed in the Methods section.

-Fig. S5J The use of the pan-PDE inhibitor IBMX to make the argument that Axl inhibition does not have an additive effect is probably not very useful. Treatment with IBMX would massively increase cAMP levels even in the absence of adrenergic stimulation. With the increase caused by IBMX being one or two orders of magnitude higher than that caused by treatment with Axl inhibitor, any additive effect by BMS would probably be masked.

We agree with the reviewer, IBMX is massively inducing intracellular cAMP levels, therefore additive BMS effects could have been masked. Therefore, we removed this experiment from the figures, as it does not add any additional information.

-Experimental procedures are described in good detail, but some more clarity is required in the description of Axl mouse gene targeting protocols:

-Description of AXL KO is somewhat confusing. "AXLKO mice were obtained by pronuclear injections of two clones (H04 and D04) of modified ES cells...". It is not clear where pronuclear injections were performed (or was it meant to be blastocyst injections?). Also, the schematic of the gene targeting vector (which should not be called 'transgenic cassette' as it creates confusion with generation of transgenic rather than gene targeted) shown in Fig. 6A is somewhat confusing with missing exons and lines connecting not continuous exons. Some more clarity needs to be provided and importantly to specify whether exons 4 and 5 were deleted (through recombination of the loxP sites in the ES cells or in the mouse through breeding with a germline Cre strain).

We apologize for the lack of clarity, we have adapted the description and the schematic.

-A reference should be provided for the AdipoQERT2Cre mouse line used in the study. Also, the efficiency of the inducible AXL inactivation in the adipose tissue is a rather important information which is missing and a western blot for AXL protein levels in the adipose tissues of conditionally deleted mice should have been shown.

A description and reference for the AdipoQERT2Cre mouse model has been added to the "Animal Work" section of the Methods. Additionally, we dissected ingWAT from iFAXLKO mice and isolated SVF and mature fractions and performed western blots analyses to show the adipocyte-specific deletion of Axl receptor.

-The term 'congenital' is normally associated with human disease. I don't think I have seen it used in the context of mouse gene targeting. 'Constitutive' knockout is the term normally used to describe this type of gene targeting.

We have changed the terminology and replaced the incorrect term “congenital” with the more fitting term “constitutive”.

Minor Comments

-Experimental Procedures. The ‘Sally Sue’ method is quoted for measuring pAXL. ‘Sally Sue’ presumably refers to an instrument manufactured by Protein Simple (not specified).

We apologize for the mistake; the information is now added to the revised manuscript, in the methods section.

-Fig. 1C histogram third bar labelled ‘scWAT’ presumably should be ‘scBAT’.

The first two bars (from the left) are referring to a different human cohort as compared to the two ones on the right, but the 3rd bar is correctly labeled as “scWAT”. We understand the confusion that this graph may create, and we have separated the columns into two separate graphs.

-Fig. 1H Dose and length of treatment with CL316,243 are not specified in the figure legend.

Dose and length of CL-treatment have been added in the figure legend.

-Fig. 2A Key of NPS doses colour coding incorrect.

The lower dosage of NPS was at the lower part of the color label, we now switched it to the correct order.

-Fig. 2A, B Although specified in the legend it would be helpful if the figure also indicated that cells are stimulated with isoproterenol in A and non-stimulated in B.

The labels “Non-stimulated iBAs” and “Isoproterenol-stimulated iBAs” have been added on Figures 2A and 2B, respectively, to make the graphs more clear.

-Fig. 2D While the two blots have different loading controls (HSP90 and tubulin) both histogram axes are labelled as ‘UCP1 intensity /HSP90’)

Y-axis label was mistakenly labeled as UCP1 intensity/HSP90 and has now been corrected to UCP1 intensity/ γ -TUBULIN.

-Fig. 4E – D What is the difference between panels E and D?

We assume that the question refers to panels E and F. Panel 4E refers to experiments in brown adipocytes whereas panel 4F refers to white adipocytes. We have labeled the panels to improve clarity. For the sake of space, we have kept only brown adipocytes in the main figure (now in Figure 4E), whereas images from white adipocytes have been moved to the supplement (Suppl. Figure 4D).

-p. 27 “AXL receptor inhibition increases intracellular cAMP levels through moderate mitigation of the Insulin/AKT/PDE pathway”. Was this meant to be “inhibition” instead of “mitigation”?

We have changed the word “mitigation” to the more appropriate term of “attenuation”.

-Fig. 5B Although mentioned in the legend, it would be clearer if Y axis label was marked as ‘isoproterenol-stimulated cAMP (pmol/mL/protein)’

The Y axis label “Iso-stimulated cAMP (pmol/mL/protein) has now been added to the figure.

-Fig. 5E Western blot data would be better presented as a histogram of average signal intensities.

The western blot data in Figure 5E have now been quantified via ImageJ and the results have been added as a graph of average signal intensities.

-Suppl. Fig.1K legend “Direct phosphodiesterase activity (absolute values are shown)”. Was it meant to be relative values?

We assume that the reviewer means Figure 5K and not Figure 1K. We showed both absolute and relative values of this direct PDE activity assay. We showed relative values in Figure 5L (highlighted both on the Y axis of the graph and in the figure legend) and absolute values in Suppl. Figure 5K (highlighted in the figure legend). For the sake of space, as many more figures have been added to address revisions, we have only kept one graph (Figure 5L).

-Suppl. Fig. 5D the time points (5 and 15 min) are not indicated in the graph.

We have now labeled the 5 and 15 min timepoints on the graph.

-Suppl. Fig. 5I What is the time point of cAMP measurement?

The timepoint was 15 min. However, we have now removed Suppl. Figure 5I from our results, as IBMX is a very broad PDE inhibitor and focused on how PDE3/4 mediate our effects, as these are more specific and relevant to adipocyte metabolism.

-Suppl. Fig. 5J Isoproterenol, BMS, IBMX doses and time point are not specified.

Suppl Figure 5J has now been removed, due to non-conclusive results based on an IBMX stimulation.

-Suppl. Fig. 5M Are cAMP levels produced by different doses of PDE4i statistically significant?

Results shown on Suppl. Figure 5M are statistically significant, we have now highlighted the statistical significance on the graph and figure legend (this graph has now been moved to Suppl. Figure 5I).

-Suppl. Fig. 6H and 6I What was the sex of mice tested?

Figure 6H is in male mice, and Figure 6I in female mice, sex of mice has been labeled in the figure legend.

-Suppl. Fig 6K This graph is probably better to present together with that of Fig. 6H.

We have rearranged the figures and have now added Suppl. Figure 6K together with Figure 6H. The new graphs are now in Figure 6G and Figure 6H.

-Suppl. Table 1 Does a dash (-) indicates complete lack of cross-reactivity or no testing?

The dash indicates complete lack of cross-reactivity. We have added this information in the figure legend.

-p35, line 733 “and lean mass (lower)”. Should read “right” instead of “lower”

Figure legend has now been adapted to reflect this change.

-p39, line 841: “in lower compared to higher conditions”. Presumably ‘temperature’ is missing.

The missing word “temperature” has now been added.

-Several graphs and schematics have too small fonts or low resolution which make them illegible.

We have now adapted all graphs and schematics to the best of our capacity and taking into consideration the available space that we have, in order to improve their resolution and increase the fonts which are not easily legible.

Reviewer #3 (Remarks to the Author):

Efthymiou et al., demonstrates a functional role of AXL inhibition in enhancing BAT thermogenesis and protection from HFD-induced obesity. They use pharmacological inhibition, siRNA and finally mice with genetic ablation of Axl to demonstrate enhanced UCP-1 and PGC-1a transcription and perform thermogenesis and HFD-weight gain studies. b-AR coupled Gs activates AC, which produces cAMP. cAMP activates PKA for thermogenesis. PDEs can negatively regulate cAMP and reduce thermogenesis. The authors postulate that GAS6-AXL dependent PI3K-PDK1-AKT axis not only excludes FOXO1 allowing PPAR γ access, but also activates PDEs and inhibits thermogenesis. Overall, the concept is interesting and surprising. AXL is more well known for its functional role in phagocytosis, anti-inflammatory signaling and promoting tumor EMT and migration/invasion.

Data is presented showing that AXL is broadly expressed including in adipose tissue. It is more prevalent in WAT relative to BAT thus giving rise to the notion that it might have a role in 'beige'-ing of WAT. Immortalized brown adipocytes were stimulated with isoproterenol. Within 4 hrs, Axl mRNA was downregulated. Protein levels reduced by 2 hr. In vivo experiments also demonstrate reduced AXL. Similar results were observed in vivo with cold exposure or b-AR agonists.

AXL inhibitors increase UCP1, PGC1a and C/EBP in brown adipocytes and enhance brown-like phenotype in mature white adipocytes. It is indicated that the other TAM receptors or MET or RON are not expressed in these cells. AXL siRNA knockdown also enhances UCP1 and PGC1a. FOXO1 is translocated. A set of signaling data is presented indicating that GAS6 enhances AXL \sim P and pAKT \sim P. It is suggested that AXL inhibition reduces insulin-induced AKT \sim P.

Finally, the authors employ Axl KO to demonstrate increased UCP1 and PGC1a, enhanced thermogenic capacity and resistance to HFD-induced weight gain. Inducible, adipocyte-specific Axl deletion was also performed. It showed results similar to Axl KO.

While the data is interesting, some improvements are required in terms of quality, rigor and controls.

Major comments

1. The inconsistent pattern in AXL immunoblots throughout the manuscript is of grave concern (see specific comments).

We apologize for this inconsistency. One of the reasons is the use of two different antibodies which were independently validated, and of which one was discontinued during the course of the study. The other reason is the different gel percentages which we had to run in order to simultaneously detect several proteins on the same membrane to allow for a direct comparison. We have now repeated several western blots and added them to the revised manuscript and explained the above-mentioned points in the manuscript. Below is a validation for both antibodies: the antibody from Cell Signaling was validated in cultured iBAs using siRNA-mediated knockdown (KD)

approaches. Axl receptor is known to have three different isoforms and – as shown in the western blot – three different bands appear in the control and are reduced in response to the siRNA-KD. This antibody that had been purchased from Cell Signaling has now been discontinued and is not commercially available. The second antibody was purchased from Abcam (Cat.No. ab215205) and has been validated in our whole-body and tissue-specific knockout (AXLKO and iFAXLKO, respectively) mouse models. As you can see, this antibody also recognizes three bands at the expected size (100-120kDa) corresponding to the three isoforms of Axl receptor. These bands do not appear in iBAT, ingWAT, and lung of AXLKO mice. In our iFAXLKO mouse model, Axl receptor deletion is driven by the Adiponectin-Cre, therefore it is expected to occur only in adipocytes. Thus, we isolated the mature fraction

from ingWAT of iFAXLKO vs. Control littermates and performed an Axl western blot (shown below). A faint band in the iFAXLKO can still be

observed, which is due to contamination of the mature adipocyte fraction with other cell types (e.g. macrophages).

2. AXL activation typically occurs when GAS6 bridges the receptor with phosphatidylserine (PtdSer). GAS6-PtdSer interaction is mediated by GLA domain of GAS. Is PtdSer-GAS6-AXL axis induced or constitutive?

We thank the reviewer for this insightful comment. To identify whether the PtdSer-Gas6-Axl axis is induced in iBAs, we stimulated mature brown adipocytes *in vitro* using either Gas6 or BMS and we subsequently measured PtdSer levels using the Annexin V assay. We did not detect any changes or differences in PtdSer levels in the presence or absence of induction of iBAs, suggesting that in our specific cell culture model of immortalized brown adipocytes, Gas6 effect may not be inducing additional bridging of the receptor with PtdSer. Together with the pharmacological inhibitor data that show downregulation of pAKT in the absence of Gas6 or in non-insulin stimulated cells, these results would suggest a constitutive activation of Axl receptor in our cultured iBAs. Results are shown in Suppl. Figure 5.

Commercially, GAS6 is available in full-length or GLA-less forms. Ideally, both forms of GAS6 should be included in the experiments to demonstrate GAS6-dependent effect on AXL. Minimally, full-length GAS6 should be used.

We thank the reviewer for this suggestion, indeed an Axl receptor activating approach will shed further light into our mechanistic studies. To that end, we employed full-length GAS6 and incorporated our findings in the revised results and figures: First, we use full-length GAS6 on figure 2 to measure gene expression in iBAs in response to the Axl receptor ligand Gas6 (Suppl. Figure 2). In addition, we utilize full-length GAS6 on figure 4 to demonstrate that pAXL (Figure 4A) and pAKT (Figure 4C and Suppl. Figure 4A) levels are increased and that this effect is dependent on Axl. Full-length Gas6 is also used on the FoxO1 localization studies (Figure 4E, 4F, and Suppl. Figure 4E). Lastly, we employed full-length GAS6 to show that Axl receptor ligand treatment is sufficient to reduce intracellular cAMP levels in isoproterenol-stimulated iBAs (Figure 5B). Overall, via employing the Axl receptor-specific ligand GAS6, we show that the effects on pAXL, pAKT, and intracellular cAMP are dependent on Axl. Even though GLA-less forms of GAS6 were not commercially available, we additionally employed an agonistic anti-AXL antibody which acts as an activating ligand. As GAS6 had minimal effects on gene expression of iBAs (Suppl. Fig. 2), we utilized this anti-AXL antibody, which is a more potent Axl ligand, as assessed by the stimulation of its effect on gene regulation (Suppl. Figure 2G, 2H, and 2I) and downstream pAKT signaling (Figure 5J). We evaluated gene expression, oxygen consumption rate (OCR) (Figure 5K), intracellular cAMP levels (Suppl. Figure 5G), and PDE activity (Figure 5L) in response to this anti-AXL antibody.

3. Conceptually, it is not apparent why Insulin-PI3K-PDK1-AKT axis is subthreshold and not sufficient for PDE activation (in the context of thermogenesis) and why is it necessary to invoke the idea of a threshold level of AKT activation that requires AXL signaling pathways? Furthermore, many RTK and other signaling pathways activate PI3K-PDK1-AKT axis. Can any of these other signals function cooperatively to exceed the threshold?

Along with the concept of distinct pools of phosphorylated AKT, it has been shown that PDEs are also present in distinct compartments. Therefore, the concept of a threshold of AKT phosphorylation could reflect either access to distinct pools of PDEs or simply activation of less molecules of PDEs (a continuum) which would or would not be sufficient to raise the levels of intracellular cAMP enough to activate crucial downstream transcription factors, such as CREB and ATF. Similarly, for PI3K, the concept of a threshold of PI3K phosphorylation could reflect either access to distinct sections of the membrane based on the local concentration of insulin receptor (IR) or other receptor tyrosine kinases (RTKs) or simply an overall lower activation/less molecules of PI3K (continuum) which would or would not be sufficient to activate downstream substrates to the aforementioned levels, we have now added a section on the discussion regarding this point.

4. One plausible explanation may be that cold induces GAS6 or PtdSer exposure along with norepinephrine release from sympathetic neurons. This should be tested.

To test this hypothesis, we stimulated mature iBAs with isoproterenol and subsequently performed the Annexin V assay, which measures PtdSer. We did not observe any significant changes between stimulated vs. non-stimulated brown adipocytes. Results are shown in Suppl. Figure 4K. Additionally, we measured circulating Gas6 levels in plasma obtained from mice that had been kept either under room-temperature conditions or under environmental cold exposure. Indeed, we observed that both isoproterenol (in vitro) and CL injections (in vivo) stimulate the expression of GAS6 in the supernatant medium or in the plasma, respectively (Figure 1J and 1K).

Specific comments

-Figure 1. The band for AXL in immunoblots is not consistent in the panels. Were different antibodies used? Especially panel H is problematic. It would be better to include the molecular weight marker in the immunoblot.

We thank the reviewer for the comment and for clarification we have added the kDa size next to the Wblot. Furthermore, we would like to refer to our answer to the Major comment 1 which addresses this issue.

-Activation of AXL leads to its cleavage at an extracellular residue and shedding of sAXL. Is AXL being cleaved following activation? What is the AXL epitope in Fig 1? Is there a corresponding increase in soluble AXL?

To measure cleavage of the extracellular part of AXL, we used iBAs in vitro and stimulated them with isoproterenol, Gas6 (Axl agonist) and BMS (Axl inhibitor) and measured circulating soluble Axl in the medium. It is very interesting that isoproterenol stimulation increases soluble Axl in the medium of the cells as this could provide a potential mechanism about the observed downregulation of Axl in response to isoproterenol or cold exposure that we have previously observed (Figure 1F and 1H). However, Gas6 reduced soluble Axl in the medium in the presence of isoproterenol, suggesting that Gas6 agonist either blocks its cleavage or it leads to its internalization, thus reducing the circulating cleaved Axl. Results are shown in Figure 1I and the data is now discussed in the revised manuscript.

-Figures 2 and 3. Strength includes use of pharmacological inhibitors and siRNA experiments.

We thank the reviewer for this comment. Indeed, using both pharmacological and siRNA approaches are crucial for conclusive results and avoidance of off-target effects.

-In figure 2 panels A and B, letters a-d are used above the graphs to likely indicate p values while * is used in the legend. Can non-normalized values be shown to better represent the isoproterenol-induced upregulation?

We apologize for these discrepancies. We have now adapted the indication of p-value on the graph so that it corresponds to the p-values as explained in the figure legend. In addition, we now present the non-normalized graph (Figure 2B) which better shows the isoproterenol-induced upregulation of Ucp1.

-Importantly, what is driving the basal AXL signaling? GAS6 should be included in these experiments to demonstrate ligand-dependent effect of AXL on transcriptional regulation.

We thank the reviewer for raising this point. We do recognize that it is indeed challenging to conclusively define what is driving the basal Axl signaling and whether it is ligand-dependent or whether there is a constitutive autophosphorylation. Based on our studies, we can conclude that Axl receptor in iBAs is constitutively active, as iBAs express and produce/secrete Gas6 into the medium (measured in Figure 1J). Similarly, we observed that GAS6 is present in the circulation in high concentrations in response to a CL injection (Figure 1K), likely leading to a constitutive activation of Axl receptor. To fully address this point, we now included GAS6 in our experiments as requested. Due to the high basal activation via Gas6, we now also employed an agonistic anti-Axl antibody as a more potent Axl receptor ligand and evaluated its effect on transcriptional regulation (results are shown in Suppl. Figure 2G, 2H, and 2I). In addition, agonistic anti-Axl antibody was employed to measure oxygen consumption rate (OCR) in response to Axl receptor activation. We show that Axl signaling activation significantly reduces OCR, as compared to an IgG control (Figure 5K). In both cases the additional activation is less prominent, which suggests that under physiological conditions Axl receptor is activated, constitutively. This point has also been added to the revised discussion.

-What may be the physiological stimulus for GAS6 production and/or PtdSer exposure?

It is likely that the cAMP/PKA/CREB-ATF signaling pathway is upstream of GAS6 production. Indeed, we show that isoproterenol-induced or CL-induced β -adrenergic stimulation increases GAS6 levels in vitro and in vivo (Figure 1). This could be part of a feedback loop system, as we also show that β -adrenergic stimulation acutely reduces Axl receptor expression and increases circulating cleaved Axl receptor (which would sequester circulating GAS6). Both

these adaptations would require the presence of increased levels of Gas6. Alternatively, such regulation could be part of a negative feedback loop in order to modulate intracellular cAMP production (β -adrenergic activation increases intracellular cAMP and in parallel stimulates GAS6 which activates Axl receptor in an autocrine/paracrine/endocrine manner, process that in turn suppresses intracellular cAMP levels). We have now added this part to the revised discussion. To test whether cold-exposure or β -adrenergic stimulation are upstream physiological stimuli for Gas6 production, we measured circulating GAS6 in the medium of iBAs in vitro (in response to isoproterenol) as well as in the serum of CL-injected mice.

-In Figure 3, similarly, GAS6 stimulation should be included for effect on UCP1 and other genes, as well as on OCR.

Based on our studies, we can conclude that Axl receptor in iBAs is constitutively active, as iBAs express and produce/secrete Gas6 into the medium (measured in Figure 1J). Similarly, we observed that GAS6 is present in the circulation in high concentrations in response to a CL injection (Figure 1K), likely leading to a constitutive activation of Axl receptor. To fully address this point, we now included GAS6 in our experiments as requested. Due to the high basal activation via Gas6, we now also employed an agonistic anti-Axl antibody as a more potent Axl receptor ligand and evaluated its effect on transcriptional regulation (results are shown in Suppl. Figure 2G, 2H, and 2I). In addition, agonistic anti-Axl antibody was employed to measure oxygen consumption rate (OCR) in response to Axl receptor activation. We show that Axl signaling activation significantly reduces OCR, as compared to an IgG control (Figure 5K). In both cases the additional activation is less prominent, which suggests that under physiological conditions Axl receptor is activated constitutively. This point has also been added to the revised discussion.

-Panel A in this figure may be confusing to the readers since normalized expression is shown. TYRO3 and MERTK absolute expression appears much reduced in comparison to AXL (higher Ct in supplementary figure).

Instead of relative mRNA expression using the ddCt method, we now show dCt values to indicate the expression levels of the gene of interest (e.g. TYRO3, MERTK, or AXL). Our data shows that TYRO3 and MERTK have much higher dCt values than AXL, suggesting a significantly lower expression. Results are shown in Suppl. Figure 1G.

-Since Axl KO and Axl f/f mice are available to the group, the pharmacological inhibition experiments can be complemented by genetic KO experiments and by showing a lack of further effect on genetically deleted lines (to rule out off-target effects). In this context, GAS6 stimulation experiments on knockout lines will add greatly to the specificity of the effects.

We thank the reviewer for this suggestion. We generated a new cohort of Axl KO mice and we treated WT and KO mice either with the AXL pharmacological inhibitor BMS-777607 or DMSO control (drug or DMSO were mixed in the high-fat diet) and followed the body weight (BW) of these mice in response to a HFD. We show that both AXL KO and BMS administration leads to a significant reduction in BW, as compared to the non-treated controls. Interestingly, the combination of AXL KO and AXL pharmacological inhibition (BMS) did not result in additional body weight loss, which suggests that the effects are indeed mediated by Axl. It should be noted that GAS6 is present in significant levels in the serum of wild-type and CL-injected mice (Fig. 1), which is the reason for the observed metabolic effects upon global or tissue specific ablation of AXL. As we now discuss in the introduction, the GAS6 knockout mouse model and its effect on adipogenesis has been previously published and it is concordant with our hypothesis and working model. GAS6 deletion does not result in body weight or fat mass differences in chow-fed mice but it leads to lower fat mass in response to a high-fat diet challenge (Maquoi et al, 2005).

-In Figure 4, BMS+GAS6 condition should be included for AKT~P. Quality of Figure 4E and F is poor and it is very difficult to clearly see FOXO1 nuclear exclusion. The major point is that AXL is driving some nuclear exclusion even in the absence of insulin. Perhaps this point can be better made by including GAS6 in the absence and presence of insulin and by including nuclear export inhibitor as positive control along with the conditions already included.

We thank the reviewer for this insightful comment. We have now replaced the FoxO1 images with better resolution images, so that the nuclear and cytoplasmic translocation is more apparent. Indeed, we observe that pharmacological Axl inhibition significantly induces nuclear FoxO1 localization. However, addition of Gas6 in the medium does not substantially induce FoxO1 translocation to the cytoplasm in the absence of insulin. We do however observe a significant increase in the amount of cytoplasmic FoxO1 upon stimulation of Gas6 in the absence of insulin. This last observation is in accordance with our previous notion that Axl receptor is constitutively activated due to the endogenous GAS6 produced/secreted by brown adipocytes, thus additional activation does not exert prominent effects.

-There are also conceptual questions about this part. The conclusion that AXL inhibition reduces insulin-stimulated AKT~P is not evident from the data. There are possibilities that are not eliminated by the experimental design. AXL is well-known to drive AKT~P when bound to its ligand GAS6. Thus, AXL inhibition would be expected to reduce AKT~P. In parallel, insulin/IGF signaling stimulates AKT~P. However, these effects can happen independently in discrete pools of AKT.

We thank the reviewer for this insightful observation. Indeed, the concept of compartmentalized pools of AKT is very interesting and could explain the effects in phosphorylation of AKT that we observe in response to AXL receptor inhibition or stimulation. Obviously, insulin stimulation would result to multiple downstream phosphorylations, and recent studies from the Kahn Lab have demonstrated the extend of these events using phosphoproteomic approaches. Since insulin is one of the strongest anabolic hormones and insulin receptor one of the main drivers of AKT phosphorylations in adipocytes, we broadly referred to the modulation of pAKT activity as insulin-stimulated signaling, as our experimental conditions included pre-starvation (FBS-depleted medium) followed by FBS-depleted medium containing only insulin (+/- inhibitors). Since we are extracting a total cell lysate, it is fairly safe to assume that a large fraction of the pAKT activity can be attributed to insulin signaling per se. Therefore, its reduction strongly points toward a crosstalk of IR-AXL signaling pathways. However, the concept of discrete AKT pools could indeed occur in the same context. Testing distinct AKT pools would go beyond the scope of the present studies. However, we have included a new part to our discussion detailing this point.

-One may also wonder that if this is simply a threshold issue, what is the actual threshold? PI3K? PDK1 activity? AKT~P?

AKT is known to impact the activity of PDEs. Additionally, AKT is a central hub for multiple metabolic pathways. We have raised the concept of the AKT threshold as a way to describe a working mechanistic model of our observations. Of course, this does not exclude the possibility that other intermediate substrates do not contribute via parallel branched signaling pathways. Therefore, it is likely that the actual "threshold" is defined by the level of activation or inactivation of AKT. At the same time, the previously described concept of distinct AKT pools points too to the direction of AKT being the hub of the actual threshold, but with its relative location and compartmentalized activation/inactivation being relevant. We have re-written the respective part of our discussion to reflect this point.

-More importantly, an explanation is required for specificity of the effect. Insulin-AKT is known to regulate PDEs via phosphorylation. Why is this axis not sufficient in the context of thermogenesis and why is it necessary to

invoke AXL? Many RTK and other signaling pathways activate PI3K-PDK1-AKT axis. Why is thermogenesis the exclusive context wherein these signaling pathways additively function to establish the threshold?

Indeed, insulin-AKT is known to regulate PDEs. We do not state that this axis is not sufficient in the context of thermogenesis. However, the necessity to invoke AXL could be derived from the fact that in biology many pathways are behaving in a “redundant” manner. It is likely that multiple signaling pathways regulating the same downstream central hubs (e.g. insulin-IR, IGF1-IGF1R, Gas6-AXL-R) are important for a tight “fine” tuning of a complex metabolic pathway. This would offer a capacity of the pathway to integrate multiple upstream signals. We have expanded our discussion to address this point.

-Figure 5 is weak. It would be extremely important to use AKT inhibitors to reverse GAS6 effects (molecular and functional).

Based on our findings, and as we discussed above, we have concluded that there is a high basal constitutive activation of Axl receptor. Therefore, we employed AKT inhibitors to mimic reversal of the GAS6 effects. We would like to highlight that chronically targeting the insulin signaling pathway with such potent Akt inhibitors may regulate multiple other pathways that would exert various effects on gene expression and would likely not correspond to the “fine-tuning” modulation that Axl inhibition exerts on Akt phosphorylation levels. Therefore, we think that these inhibitors should only be used as tools to demonstrate acute effects and related molecular mechanisms (e.g. on insulin-signaling substrates phosphorylation levels or cAMP levels) and not in a chronic manner. To that end, and as suggested by the reviewer, we used AKT inhibitors and activators, PI3K inhibitors, as well as PTEN inhibitors and measured their effect on intracellular cAMP levels (Fig. 5G and 5H). Akt inhibitors and PI3K inhibitors indeed increase cAMP levels, whereas Akt activators and PTEN inhibitors exert the opposite effect and suppress elevation of cAMP levels (Fig. 5G and 5H).

For the chronic experiments we employed PKA inhibitors that will provide more conclusive results, as it would not affect all insulin dependent signaling cascades. To that end, we chronically treated cells with the well characterized PKA inhibitor H89 and we indeed show that it abrogates the effect of Axl inhibitor on expression of Ucp1 and Pgc1a, suggesting that this pathway is essential and mediates its effect. In addition to demonstrate the molecular mode of action, we inhibited ATF2 which is a known TF downstream of the cAMP signaling pathway and we could indeed abrogate the effects of Axl inhibitors on gene expression of Ucp1 and Pgc1a (Figure 5N). We have now revised our results and discussion sections to address these points in further detail and reflect these changes.

-The use of GAS6 here is useful. It would be best to be consistent in using GAS6 throughout the experiments reported in the manuscript.

We agree with the reviewer that including an AXL receptor activation approach would be useful for the interpretation of a part of our conclusions. In fact, throughout the manuscript we have included several experiments where we employ either the endogenous ligand GAS6 or the agonistic anti-AXL antibody (AB) as tools to activate AXL receptor, and have measured effects on gene expression, AKT phosphorylation, FoxO1 localization, cAMP levels, OCR, and PDE activity as described above.

-The PTEN inhibitors are known to be non-selective (PMID: 25446882). Constitutively active PI3K should be used to validate these findings. Similarly, constitutively active and kinase inactive AKT constructs should be used.

Overexpressing constitutively active and kinase inactive constructs via virus-mediated tools would have the drawback that the lag until they are fully expressed renders them irrelevant to the window of the “activation assay” that we have established in our cultured brown adipocytes (days 5-8 of differentiation). We do agree with

the reviewer that PTEN inhibitors – similar to many small-molecule compounds – may exert non-selective effect. However, in our experiments we employ PTEN inhibitors as tools to modulate pAKT levels, which we achieve with low dosages of these compounds (e.g. VO-Ohipic is highly potent and reverses the effects of BMS on pAKT at a concentration of 100nM, and SF1670 has been described as a highly specific PTEN inhibitor), thus reducing the likelihood of non-selective effects. In addition, these compounds have been used in multiple publications, supporting the fact that they exert their major effects via PTEN. To further address this comment, we employed pharmacological approaches with AKT inhibitors and activators, as well as PI3K inhibitors. Results are shown in Figure 5G and 5H. These AKT inhibitors and activators are extremely well established in the literature, show high potency and selectivity, and have been employed in multiple publications. Together this suggests that the majority of the effect of AXL inhibition is mediated by AKT.

-Again, the AXL band is inconsistent in panel I.

Please, refer to our response above (major comment #1) where we provide a detailed explanation for the discrepancies regarding the AXL bands in western blots as well as a detailed validation of the antibodies used in our studies.

-Figure 6 is a highly significant and important figure. However, there are concerns. Why is a AXL band present in lung of KO with CST antibody? Was Axl mRNA levels tested in the KO and iKO?

The AXL band in the lung of KO is attributed to background/noise of the Wblot, due to poor quality of the antibody used. We repeated the Wblot with a new antibody, and as we show in Figure 6A, AXL is not present in any tissue in the AXL KO mouse model. Additionally, we confirm/validate that AXL is expressed in higher levels in WAT vs. BAT. Additionally, we performed an AXL Wblot in the mature adipocyte fraction of the iFAXLKO mice (as deletion is driven by Adiponectin) and we confirm that AXL is successfully deleted in the iFAXLKO mouse model, specifically in adipocytes (Figure 6K). Please, also refer to our response above (major comment #1) where we provide a detailed explanation for the discrepancies regarding the AXL bands in western blots as well as a detailed validation of the antibodies used in our studies.

-There is a published Axl CR/CR mouse (PMID: 33529176) where AXL signaling should be enhanced. It is tempting to speculate what would happen with regards to this mouse and thermogenesis.

We thank the reviewer for bringing this very interesting study to our attention. Even though, in our present studies we have majorly focused on the brown-fat mediated effects of AXL receptor deletion, our model does not entirely rule out that beneficial effects of the AXLKO mice could be attributed to immune cells too. In fact, our thermoneutrality experiments point towards the direction of brown-fat mediated beneficial effects. However, activation of macrophages has been shown to play a role in brown fat activation. It is tempting to speculate that upon a whole-body AXL KO, macrophages are activated towards the beneficial M2 phenotype, which has been shown to induced brown fat activation. Therefore, our phenotype could partially be attributed to macrophage-mediated effects and not purely on adipocyte / adipogenic progenitor – mediated effects. We have included a relevant part in our discussion, to include the aforementioned speculations.

-In Supplementary Figure 3D, the names of most differentially genes could be included.

We have now annotated the names of the most differentially expressed genes in Figure 3D.

-Since the knockout mouse lines are novel, they should be thoroughly characterized for genomic deletion (including sequencing), mRNA and for protein expression. Transcriptional profiling would also be useful.

As suggested, we have performed bulk RNA-Sequencing in iBAT from AXLKO vs WT mice (data shown in Figure 6J and Suppl. Figure 6J), as well as in iBAT from iFAXLKO vs. Cre- WT mice (data shown in Suppl. Figure 6K and 6L). Additionally, we phenotyped the two AXLKO mouse models in great detail and included this data in the revised version of the manuscript.

REVIEWERS' COMMENTS

Reviewer #1 (Remarks to the Author):

The authors addressed all my experimental points. The new data clearly improved the manuscript. I would stress that the new feedback loop should be clarified to the readers by modifying the scheme.

Reviewer #2 (Remarks to the Author):

The revised manuscript by Efthymiou et al. includes a significant body of additional experimental evidence that supports the conclusions of the original manuscript and provides some further mechanistic insights to the reported phenotypes. In my opinion, the most important additions are:

-The demonstration of a Gas6 autocrine/paracrine mechanism of basal activation of the Axl signalling pathway in adipocytes (data shown in Fig. 1I-K), which explains the effects of inhibition of the Axl receptor.

-The role of Foxo1 in transcriptional regulation of UCP-1 and PGC1a induction through Axl inhibition is substantiated from the evidence provided using a chemical inhibitor and RNAi for Foxo1 (data shown in Fig. 4G-H).

-the additional metabolic phenotyping of global and adipose tissue-specific Axl KO mice, which demonstrates improved insulin sensitivity and glucose tolerance in the latter (data shown in Fig. 6 and S6).

In my view, the revised manuscript addresses the criticisms raised during the original review more than adequately and offers further significant insights in the mechanisms underlying the reported phenotypes. It is an impressive body of very interesting work, which makes a significant contribution to the understanding of the biology of the adipose tissue and it has obvious potential therapeutic implications.

Minor comments: Although the authors have made a strong effort to improve the presentation of the manuscript, a few oversights remain, and it requires some further proofreading and polishing. For example: the term 'congenital' instead of 'constitutive' is still used in the abstract; 'pronuclear' instead of blastocyst injection of ES cells in materials and methods, Fig.4D legend 'pharmacokinetics' instead of 'time-course'; Fig. 7, the arrow from PDE to cAMP should be an inhibitory one (this obviously is not an exhaustive list).

Reviewer #3 (Remarks to the Author):

Efthymiou et al. have significantly revised their manuscript in response to the suggestions by the Reviewers. For example, they have better characterized the iFAXLKO mice including bulk RNAseq in iBAT adipose depots. The metabolic tests are also an improvement. They also use a FOXO1 inhibitor to reverse the gene expression changes induced by AXL inhibition. As far as my previous comments – the authors performed new experiments and adequately addressed by concerns. They also provide reasonable justifications and additional clarifications for the antibodies used and inconsistencies in the immunoblots. These are acceptable justifications. Thus, the rigor of the manuscript is enhanced.

The authors also include GAS6 to stimulate AXL but report that autocrine GAS6 makes this experiment redundant. PtdSer levels, as detected by Annexin V, are unchanged. A question still remains as to how the GLA domain of GAS6 is relevant. One would still expect the mechanism of GAS6-dependent constitutive AXL activation to require PtdSer bridging by GAS6 gamma-carboxylated GLA domain. Thus, my prior requests for comparing GLA-GAS6 versus GLA-less GAS6. Perhaps, blocking PtdSer access to GAS6 by including free Annexin V may be a relevant experiment but I leave it to the discretion of the authors.

The finding that AXL activation prevents its cleavage is surprising. This goes against the dogma of TAM RTK activation and proteolysis but I would take the data at face value.

Minor point – I referred to the AXL CR/CR mice not to indicate to role of macrophages – albeit important and interesting – but as AXL gain-of-function (and presumably, a worse outcome) to contrast with the AXL loss-of-function/inhibition and favorable outcome models demonstrated here in.

Collectively, the finding that AXL inhibition/loss-of-function enhances thermogenic capacity of brown and white adipocytes and protects against high fat diet-induced obesity is very interesting and significant. I have no additional criticisms.

Response Letter to the Reviewers

Reviewer #1 (Remarks to the Author):

The authors addressed all my experimental points. The new data clearly improved the manuscript. I would stress that the new feedback loop should be clarified to the readers by modifying the scheme.

We thank the reviewer for the comment. We have now clarified the new feedback loop by modifying the scheme.

Reviewer #2 (Remarks to the Author):

The revised manuscript by Efthymiou et al. includes a significant body of additional experimental evidence that supports the conclusions of the original manuscript and provides some further mechanistic insights to the reported phenotypes. In my opinion, the most important additions are:

-The demonstration of a Gas6 autocrine/paracrine mechanism of basal activation of the Axl signalling pathway in adipocytes (data shown in Fig. 1I-K), which explains the effects of inhibition of the Axl receptor.

-The role of Foxo1 in transcriptional regulation of UCP-1 and PGC1a induction through Axl inhibition is substantiated from the evidence provided using a chemical inhibitor and RNAi for Foxo1 (data shown in Fig. 4G-H).

-the additional metabolic phenotyping of global and adipose tissue-specific Axl KO mice, which demonstrates improved insulin sensitivity and glucose tolerance in the latter (data shown in Fig. 6 and S6).

In my view, the revised manuscript addresses the criticisms raised during the original review more than adequately and offers further significant insights in the mechanisms underlying the reported phenotypes. It is an impressive body of very interesting work, which makes a significant contribution to the understanding of the biology of the adipose tissue and it has obvious potential therapeutic implications.

We thank the reviewer for the comments and for confirming that we have more than adequately addressed the criticisms raised.

Minor comments: Although the authors have made a strong effort to improve the presentation of the manuscript, a few oversights remain, and it requires some further proofreading and polishing. For example: the term 'congenital' instead of 'constitutive' is still used in the abstract; 'pronuclear' instead of 'blastocyst injection of ES cells' in materials and methods, Fig.4D legend 'pharmacokinetics' instead of 'time-course'; Fig. 7, the arrow from PDE to cAMP should be an inhibitory one (this obviously is not an exhaustive list).

We thank the reviewer for bringing these details to our attention. We have replaced the term "congenital" with the term "constitutive" in the abstract, the term "pronuclear" with the term "blastocyst injection of ES cells" in the methods section, and the term "pharmacokinetics" with the term "time-course" in Figure Legend 4D. On Figure 7, we have replaced the arrow from PDE to cAMP with an inhibitory sign.

Reviewer #3 (Remarks to the Author):

Efthymiou et al. have significantly revised their manuscript in response to the suggestions by the Reviewers. For example, they have better characterized the iFAXLKO mice including bulk RNAseq in iBAT adipose depots. The metabolic tests are also an improvement. They also use a FOXO1 inhibitor to reverse the gene

expression changes induced by AXL inhibition. As far as my previous comments – the authors performed new experiments and adequately addressed by concerns. They also provide reasonable justifications and additional clarifications for the antibodies used and inconsistencies in the immunoblots. These are acceptable justifications. Thus, the rigor of the manuscript is enhanced.

We thank the reviewer for stating that we have significantly revised the manuscript, that our justifications are reasonable, and that the rigor of the manuscript is enhanced.

The authors also include GAS6 to stimulate AXL but report that autocrine GAS6 makes this experiment redundant. PtdSer levels, as detected by Annexin V, are unchanged. A question still remains as to how the GLA domain of GAS6 is relevant. One would still expect the mechanism of GAS6-dependent constitutive AXL activation to require PtdSer bridging by GAS6 gamma-carboxylated GLA domain. Thus, my prior requests for comparing GLA-GAS6 versus GLA-less GAS6. Perhaps, blocking PtdSer access to GAS6 by including free Annexin V may be a relevant experiment but I leave it to the discretion of the authors.

Indeed, how the GLA domain of GAS6 is relevant remains an open question. As the main focus of our studies is the effects of the inhibition of AXL receptor, we think that this important point needs to be addressed as part of future studies.

The finding that AXL activation prevents its cleavage is surprising. This goes against the dogma of TAM RTK activation and proteolysis but I would take the data at face value.

Indeed, this is a surprising finding and future studies are required to better understand how this is regulated and its biological significance.

Minor point – I referred to the AXL CR/CR mice not to indicate to role of macrophages – albeit important and interesting – but as AXL gain-of-function (and presumably, a worse outcome) to contrast with the AXL loss-of-function/inhibition and favorable outcome models demonstrated here in.

We thank the reviewer for this comment, we have added a part in the discussion to highlight the interesting phenotype of the cleavage-resistant AXL CR/CR mouse model and put it in perspective to our observations.

Collectively, the finding that AXL inhibition/loss-of-function enhances thermogenic capacity of brown and white adipocytes and protects against high fat diet-induced obesity is very interesting and significant. I have no additional criticisms.

We thank the reviewer for finding our studies interesting and significant.